# A NEURAL MATERIAL POINT METHOD FOR PARTICLE-BASED EMULATION

## ABSTRACT

Mesh-free Lagrangian methods are widely used for simulating fluids, solids, and their complex interactions due to their ability to handle large deformations and topological changes. These physics simulators, however, require substantial computational resources for accurate simulations. To address these issues, deep learning emulators promise faster and scalable simulations, yet they often remain expensive and difficult to train, limiting their practical use. Inspired by the Material Point Method (MPM), we present NeuralMPM, a neural emulation framework for particle-based simulations. NeuralMPM interpolates Lagrangian particles onto a fixed-size grid, computes updates on grid nodes using image-to-image neural networks, and interpolates back to the particles. Similarly to MPM, NeuralMPM benefits from the regular voxelized representation to simplify the computation of the state dynamics, while avoiding the drawbacks of mesh-based Eulerian methods. We demonstrate the advantages of NeuralMPM on 6 datasets, including fluid dynamics and fluid-solid interactions simulated with MPM and Smoothed Particles Hydrodynamics (SPH). Compared to GNS and DMCF, NeuralMPM reduces training time from 10 days to 15 hours, memory consumption by 10x-100x, and increases inference speed by 5x-10x, while achieving comparable or superior long-term accuracy, making it a promising approach for practical forward and inverse problems. A project page is available at [URL].

## 1 INTRODUCTION

The Navier-Stokes equations describe the time evolution of fluids and their interactions with solid materials. As analytical solutions rarely exist, numerical methods are required to approximate the solutions. On the one hand, Eulerian methods discretize the fluid domain on a fixed grid, simplifying the computation of the dynamics, but requiring high-resolution meshes to solve small-scale details in the flow. Lagrangian methods, on the other hand, represent the fluid as virtual moving particles defining the system's state, hence maintaining a high level of detail in regions of high density. While effective at handling deformations and topological changes (Monaghan, 2012), Lagrangian methods struggle with collisions and interactions with rigid objects (Lind et al., 2020; Vacondio et al., 2021).

Regardless of the discretization strategy, large-scale high-resolution numerical simulations are computationally expensive, limiting their practical use in downstream tasks such as forecasting, inverse problems, or computational design. To address these issues, deep learning emulators have shown promise in accelerating simulations by learning an emulator model that can predict the system's state at a fraction of the cost. Next to their speed, neural emulators also have the strategic advantage of being differentiable, enabling their use in inverse problems and optimization tasks (Allen et al., 2022; Forte et al., 2022; Zhao et al., 2022). Moreover, they have the potential to be learned directly from real data, bypassing the costly and resource-intensive process of building a simulator (He et al., 2019; Jumper et al., 2021; Lam et al., 2023; Lemos et al., 2023; Pfaff et al., 2021). In this direction, particle-based neural emulators (Prantl et al., 2022; Sanchez-Gonzalez et al., 2020; Ummenhofer et al., 2020) have seen success in accurately simulating fluids and generalizing to unseen environments. These emulators, however, suffer from the same issues as traditional Lagrangian methods, with collisions and interactions with rigid objects being particularly challenging. These emulators may also require long training and inference times, limiting their practical use.

Taking inspiration from the hybrid Material Point Method (MPM) (Nguyen et al., 2023; Sulsky et al., 1993) that combines the strengths of both Eulerian and Lagrangian methods, we introduce NeuralMPM, a neural emulation framework for particle-based simulations. As in MPM, NeuralMPM maintains Lagrangian particles to represent the system's state but models the system dynamics on voxelized representations. In this way, NeuralMPM benefits from a regular grid structure to simplify the computation of the state dynamics but avoids the drawbacks of mesh-based Eulerian methods. By interpolating the particles onto a fixed-size grid, it also bypasses the need to perform an expensive neighbor search at every timestep, substituting it with two interpolation steps based on cheap voxelization (Xu et al., 2021). By defining the system dynamics on a grid, NeuralMPM can also leverage well-established grid-to-grid neural architectures. The resulting inductive bias allows the model to more easily process the global and local structures of the point cloud, instead of having to discover them, and frees capacity for learning the dynamics of the system represented by the grid. Compared to previous data-driven approaches (Prantl et al., 2022; Sanchez-Gonzalez et al., 2020; Ummenhofer et al., 2020), these improvements reduce the training time from days to hours, while achieving higher or comparable accuracy.

## 2 COMPUTATIONAL FLUID DYNAMICS

Computational fluid dynamics simulations can be classified into two broad categories, Eulerian and Lagrangian, depending on the discretization of the fluid (Rakhsha et al., 2021). In Eulerian simulations, the domain is discretized with a mesh, with state variables $u_i^t$ (such as mass or momentum) maintained at each mesh point $i$. Well-known examples of Eulerian simulations are the finite difference method, where the domain is divided into a uniform regular grid (also called an Eulerian grid), and the finite element method, where the domain is divided into regions, or elements, that may have different shapes and density, allowing to increase the resolution in only some areas of the domain (Iserles, 2008; Morton & Mayers, 2005). Lagrangian simulations, on the other hand, discretize the fluid as a set of virtual moving particles $\{p_i^t, u_i^t\}_{i=1}^N$, each described by its position $p_i^t$ and state variables $u_i^t$ that include the particle velocity $v_i^t$. To simulate the fluid, the particles move according to the dynamics of the system, producing a new set of particles $\{p_i^{t+1}, u_i^{t+1}\}_{i=1}^N$ at each timestep. Simulations in Lagrangian coordinates are particularly useful when the fluid is highly deformable, as the particles can move freely and adapt to the fluid's shape. Among Lagrangian methods, Smoothed Particle Hydrodynamics (SPH) is one of the most popular, where the fluid is represented by a set of particles that interact with each other through a kernel function that smooths the interactions.

Hybrid Eulerian-Lagrangian methods combine the strengths of both frameworks. Like Lagrangian methods, they carry the system state information via particles, thereby automatically adjusting the resolution to the local density of the system. By using a regular grid, however, they simplify gradient computation, make entity contact detection easier, and prevent cracks from propagating only along the mesh. Among hybrid methods, the Material Point Method has gained popularity for its ability to handle large deformations and topological changes. MPM combines a regular Eulerian grid with moving Lagrangian particles. It does so in four main steps: (1) the quantities carried by the particles are interpolated onto a regular grid $G^t = \mathbf{p2g}(\{p_i^t, u_i^t\})$ using a particle-to-grid (**p2g**) function, (2) the equations of motion are solved on the grid, where derivatives and other quantities are easier to compute, resulting in a new grid state $G^{t+1} = f(G^t)$, (3) the resulting dynamics are interpolated back onto the particles as $\{u_i^{t+1}\} = \mathbf{g2p}(G^{t+1}, \{p_i^t\})$, using a grid-to-particle (**g2p**) function, (4) the positions of the particles are updated by computing particle-wise velocities and using an appropriate integrator, such as Euler, i.e., $p_i^{t+1} = p_i^t + \Delta t v_i^{t+1}$. The grid values are then reset for the next step. MPM has been used in soft tissue simulations (Ionescu et al., 2005), in molecular dynamics (Lu et al., 2006), in astrophysics (Li & Liu, 2002), in fluid-membrane interactions (York II et al., 2000), and in simulating cracks (Daphalapurkar et al., 2007) and landslides (Llano Serna et al., 2015). MPM is also widely used in the animation industry, perhaps most notably in Disney's 2013 film Frozen (Stomakhin et al., 2013), where it was used to simulate snow.

Notwithstanding the success of numerical simulators, they remain expensive, slow, and, most of the time, non-differentiable. In recent years, differentiable neural emulators have shown great promise in accelerating fluid simulations, most notably in a series of works to emulate SPH simulations in a fully data-driven manner. Graph network-based simulators (GNS) (Sanchez-Gonzalez et al., 2020) use a graph neural network (GNN) and a graph built from the local neighborhood of the particles to predict the acceleration of the system. The approach requires building a graph out

of the point cloud at every timestep to obtain structural information about the cloud, which is an expensive operation. In addition, the GNN needs to extract global information from its nodes, which is only possible with a high number of message-passing steps, resulting in a large computational graph and long training and inference times. This large computational graph, along with repeated construction, makes fully autoregressive training over long rollouts impractical, as the gradients need to backpropagate all the way back to the first step. Cheaper strategies exist, like the push-forward trick (Brandstetter et al., 2022b), but they have been shown to be inferior to fully backpropagating through trajectories (List et al., 2024; Sharabi & Louppe, 2023). As autoregressive training is not available, the stability of the learned dynamics can be compromised, making the model prone to diverging or oscillating. Noise injection training strategies can be used to increase the stability of the rollouts, but the magnitude of the noise becomes a critical parameter. Han et al. (2022) introduce improvements to GNS to make them subequivariant to certain transformations. They show increased accuracy on simulations involving solid objects. An alternative approach is the continuous convolution (CConv) (Ummenhofer et al., 2020; Winchenbach & Thuerey, 2024), an extension of convolutional networks to point clouds. In this method, a convolutional kernel is applied to each particle by interpolating the values of the kernel at the positions of its neighbors, which are found via spatial hashing on GPU, a cheaper alternative to tree-based searches that allows for autoregressive training. In (Prantl et al., 2022), Deep Momentum Conserving Fluids (DMCF) build upon CConv to design a momentum-conserving architecture. Nevertheless, to account for long-range interactions, the authors introduce different branches, with different receptive fields, into their network. The number of branches, and their hyperparameters, need to be tuned to capture global dependencies, leading to long training times even with optimized CUDA kernels. Finally, Zhang et al. (2020), propose an approach that uses nearest neighbors to construct the local features of each particle. Those local features are then averaged onto a regular grid. Like GNS, this method suffers from the need to repeat the neighbor search at every simulation timestep. Ultimately, the performance of point cloud-based simulators is tightly linked to the method used to process the spatial structure of the cloud. Brute force neighbour search is $\mathcal{O}(N^2)$, K-d trees are $\mathcal{O}(N \log N)$, and voxelization and hashing are $\mathcal{O}(N)$ (Hastings & Mesit, 2005; Xu et al., 2021).

An alternative to data-driven modeling is the use of hybrid models, where parts of a classical solver are replaced with learned components. For instance, Yin et al. (2021) employ a neural network to learn unknown physics, which is then integrated into a simulator. Similarly, Li et al. (2024) use a neural network to bypass computational bottlenecks in MPM simulators, while Ma et al. (2023) learn general constitutive laws, allowing for one-shot trajectory learning. These approaches achieve impressive results by leveraging extensive physics knowledge, but this reliance also limits their applicability. Hybrid models inherit both the strengths and weaknesses of classical and ML methods.

## 3 NEURALMPM

We consider a Lagrangian system evolving in time and defined by the positions $p_i^t$ and velocities $v_i^t$ of a set of $N$ particles $i = 1, ..., N$. We denote with $P^t$ and $V^t$ the set of positions and velocities of all particles at time $t$ and with $S^t = (P^t, V^t)$ the full state of the system. In a more complex setting, the state of the system can include other local properties, such as pressure or elastic stiffness of materials, and global properties, such as an external force. In this work, for simplicity, we let the network learn the relevant simulation parameters implicitly. The evolution of the particles is described by a function $f$ mapping the current state of the system to its next state $S^{t+1} = f(S^t)$. Given a starting system $S^0 = (P^0, V^0)$, its full trajectory, or rollout, is denoted by $S^{1:T}$. Our goal is to build an emulator $\hat{f}_\theta(\cdot)$ capable of predicting a full rollout $\hat{f}_\theta^{1:T}(S^0)$ of $T$ timesteps from the initial state $S^0$. Following MPM, NeuralMPM operates in four steps, as illustrated in Figure 1:

**Step 1: Voxelization.** Using the particle positions $P^t$, the velocities $V^t$ are interpolated onto a regular fixed-size grid. This interpolation is performed through *voxelization*, which divides the domain into regular volumes (voxels). Each grid node is identified as the center of a voxel (e.g., square in 2D) in the domain, and the velocities of the particles in the voxel are averaged to give the node's velocity. Similarly, the density is computed as the normalized number of particles in the voxel. This results in the grid tensor $G^t$ that contains the grid velocities $V_g^t$ and density $D_g^t$.

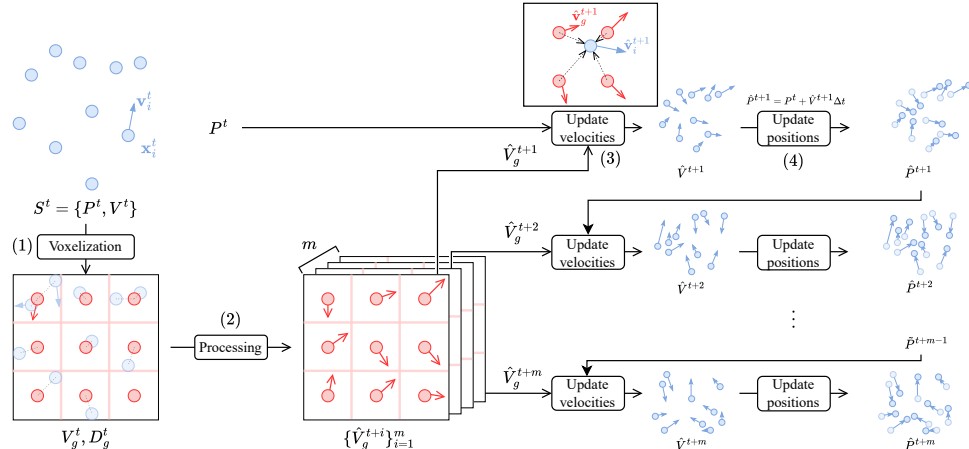

Figure 1: NeuralMPM works in 4 steps. (1) The positions $P^t$ and velocities $V^t$ of the particles are used to compute the velocity $V_g^t$ and density $D_g^t$ of each grid node through voxelization. (2) From this grid, the processor neural network predicts the grid velocities at the next $m$ timesteps. The next $m$ positions are computed iteratively by (3) performing bilinear interpolation of the predicted velocities onto the previous positions and (4) updating the positions using the predicted velocities.

**Step 2: Processing.** Taking advantage of the regular grid representation of the cloud, the grid velocities $\{\hat{V}^i\}_{i=t+1}^{t+m}$ of the next $m$ timesteps are predicted using a neural network. We chose a U-Net (Ronneberger et al., 2015) as it is a well-established image-to-image model, known to perform well in various tasks, including physical applications. The combination of kernels applied with different receptive fields (from smaller to larger) allows the U-Net to efficiently extract both local and global information. Nonetheless, any grid-to-grid architecture could be used. We experiment with the FNO (Li et al., 2021) architecture in Appendix B and find it to underperform, leading us to keep the U-Net. A fully convolutional U-Net and an FNO have the additional advantage of being able to generalize to different domain shapes, a desirable property (Section 4.3).

**Step 3: Update of particle velocities.** The predicted velocities $\hat{V}^{t+1}$ at the next timestep are then interpolated back to the particle level onto the positions $P^t$ using bilinear interpolation. The velocity of each particle is computed as a weighted average of the four surrounding grid velocities, based on its Euclidean distance to each of them.

**Step 4: Update of particle positions.** Finally, the positions of the particles are updated with Euler integration using the next velocities and known current positions of the particles, that is $\hat{P}^{t+1} = P^t + \Delta t \hat{V}^{t+1}$. Steps 3 and 4 are performed $m$ times to compute the next $m$ positions from the set of grid velocities computed at step 2.

Additional features of the individual particles can be included in the grid tensor $G^t$ by interpolating them in the same way as the velocities. Local, such as boundary conditions, or global, such as gravity or external forces, features are represented as grid channels. For simulations with multiple types of particles, the features of each material are interpolated independently and stacked as channels in $G_t$.

NeuralMPM is trained end-to-end on a set of trajectories $S^{0:T}$ to minimize the mean squared error $||P^{t+1} - \hat{P}_\theta^{t+1}(S^t)||_2^2$ between the ground-truth and predicted next positions of the particles. At inference time, the model is exposed to much longer sequences, which requires carefully stabilizing the rollout procedure to prevent the accumulation of large errors over time. To address this, we first make use of autoregressive training (Prantl et al., 2022; Ummenhofer et al., 2020), where the model is unrolled $K$ times on its own predictions, producing a sequence of $\hat{S}^k = \hat{f}_\theta(\hat{S}^{k-1})$ for $k = 1, ..., K$ and initial input $\hat{S}^0 = S^0$, before backpropagating the error through the entire rollout. Unlike more costly methods that require alternative stabilization strategies, such as noise injection (Sanchez-Gonzalez et al., 2020), NeuralMPM's efficiency makes autoregressive training

possible. Nevertheless, to further stabilize the training, we couple autoregressive training with time bundling (Brandstetter et al., 2022b), resulting in a training strategy where the model predicts $m$ steps $\hat{S}^{1:m}$ at once from a single initial state, inside an outer autoregressive loop of $K$ steps of length $m$. We show in Section 4 that this training strategy leads to more accurate rollouts.

## 4 EXPERIMENTS

We conduct a series of experiments to demonstrate the accuracy, speed, and generalization capabilities of NeuralMPM. Specifically, we examine its robustness to hyperparameter and architectural choices through an ablation study (4.1). We compare NeuralMPM to GNS and DMCF in terms of accuracy, training time, convergence, and inference speed (4.2). We also evaluate the generalization capabilities of NeuralMPM (4.3) and illustrate how its differentiability can be leveraged to solve an inverse design problem (4.4). Through these experiments, we demonstrate that NeuralMPM is a flexible, accurate, and fast method for emulating complex particle-based simulations. The baselines established by Winchenbach & Thuerey (2024) and hybrid simulators (Li et al., 2024; Ma et al., 2023) have promising results. However, we do not compare against them as they either use different benchmarks or are specifically tailored for certain physical domains, requiring material-specific knowledge. In contrast, NeuralMPM, like GNS and DMCF, requires only particle positions without being restricted to any particular domain.

**Data.**  We consider 6 datasets with variable sequence lengths, numbers of particles, and materials. The first three datasets, WATERRAMPS, SANDRAMPS, and GOOP, contain a single material, water, sand, and goop, respectively, with different material properties. The first two datasets contain random ramp obstacles to challenge the model's generalization capacity. The fourth dataset, MULTI-MATERIAL, mixes the three materials together in the same simulations. These four datasets are taken from Sanchez-Gonzalez et al. (2020) and were simulated using the Taichi-MPM simulator (Hu et al., 2018b). They each contain 1000 trajectories for training and 30 (GOOP) or 100 (WATERRAMPS, SANDRAMPS, MULTIMATERIAL) for validation and testing. The fifth dataset, DAM BREAK 2D, was generated using SPH and contains 50 trajectories for learning, and 25 for validation and testing. The last dataset, VARIABLEGRAVITY, was also generated using Taichi-MPM. It consists of simulations with variable gravity of a water-like material, and contains 1000 trajectories for training and 100 for validation and testing.

**Protocol.**  NeuralMPM is trained on trajectories with varying initial conditions and number of particles. The training batches are sampled randomly in time and across sequences. We use Adam (Kingma & Ba, 2014) with the following learning rate schedule: a linear warm-up over 100 steps from $10^{-5}$ to $10^{-3}$, 900 steps at $10^{-3}$, then a cosine annealing (Loshchilov & Hutter, 2017) for $100,000$ iterations. We use a batch size of 128, $K = 4$ autoregressive steps per iteration, bundle $m = 8$ timesteps per model call (resulting in 24 predicted states), and a grid size of $64 \times 64$. For most of our experiments, we use a U-Net (Ronneberger et al., 2015) with three downsampling blocks with a factor of 2, 64 hidden channels, a kernel size of 3, and MLPs with three hidden layers of size 64 for pixel-wise encoding and decoding into a latent space. For a fair comparison, we ran training and inference for NeuralMPM, DMCF, and GNS on the exact same hardware. GNS and DMCF were trained until convergence (a maximum of 120 and 240 hours, respectively), while NeuralMPM required 20 hours or less to converge. For WATERRAMPS, SANDRAMPS, GOOP, and MULTIMATERIAL, we use the same parameters as those reported by authors. We hyperparameter search DMCF for DAM BREAK 2D and both GNS and DMCF for VARIABLEGRAVITY and report the best performance obtained for a budget of 60 GPU-days per dataset. Further details on training can be found in Appendix A.

### 4.1 ABLATION STUDY

To study the robustness of NeuralMPM to hyperparameter and architectural choices, we start with the default architecture and hyperparameters and ablate its components individually to examine their impact on performance. We vary the number $K$ of autoregressive steps with and without noise, the number of bundled timesteps $m$ predicted by a single model call, and the depth and number of hidden channels of the network. We also investigate adding noise to stabilize rollouts, either directly to the particles' positions or to the grid-level representation after voxelization.

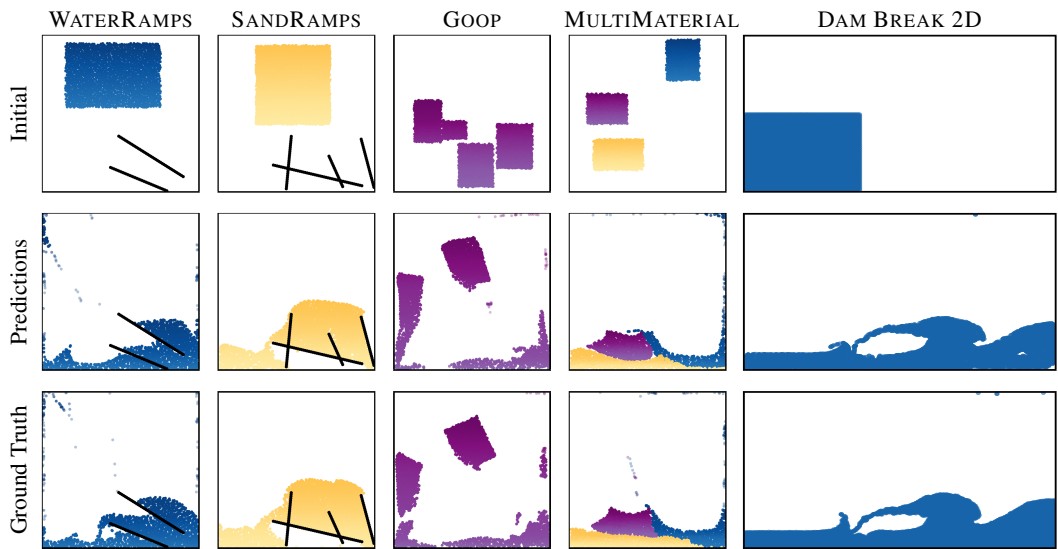

Figure 2: **Example snapshots.** We train and evaluate NeuralMPM on WATERRAMPS, SAN-DRAMPS and GOOP, each consisting of a single material, on MULTIMATERIAL that mixes water, sand and goop, and on DAM BREAK 2D, a rectangular-shaped SPH dataset. NeuralMPM is able to learn various kinds of materials, their interactions, and their interactions with solid obstacles. Despite being inspired by MPM, it is not limited to data showing MPM-like behaviour.

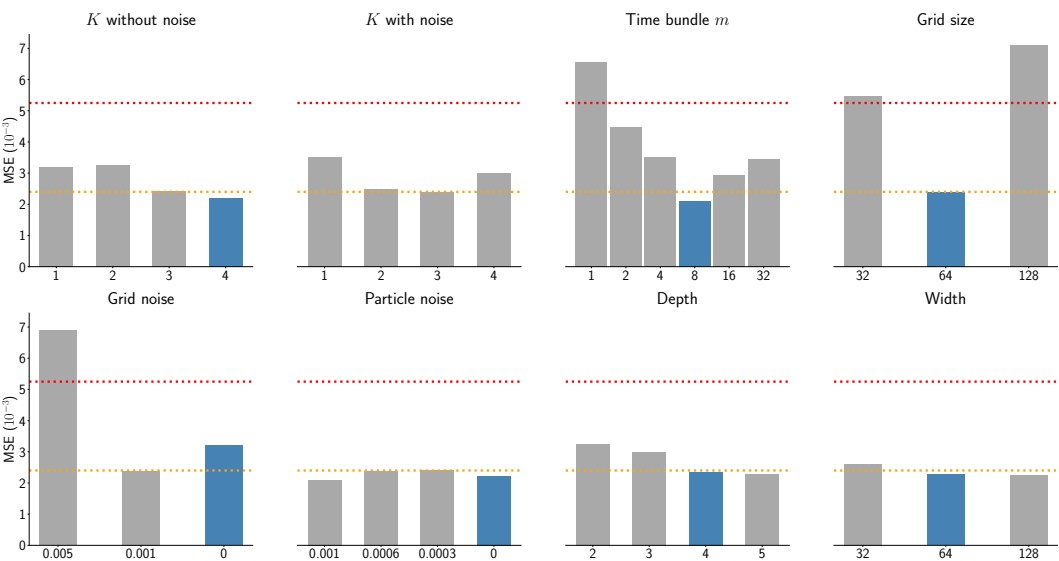

Figure 3: **Ablation results.** Mean squared error (MSE) of full rollouts on unseen test data for GOOP. The default parameters are in blue. The dotted orange line ($2.4 \times 10^{-3}$) indicates the MSE we obtained for GNS after 240 hours (20M training steps). The dotted red line is the MSE for DMCF after the same amount of time ($5.25 \times 10^{-3}$). NeuralMPM is robust to hyperparameter changes, with the biggest effects coming from the number of timesteps bundled together ($m$) and grid noise. For a rollout of length $T$, the model is called $T/m$ times, meaning lower values of $m$ require maintaining stability for longer. Autoregressive training coupled with time bundling is sufficient to stabilize the model, eliminating the need for noise injection. Although GNS reportedly outperforms NeuralMPM by a small margin, these results could not be reproduced in our experiments.

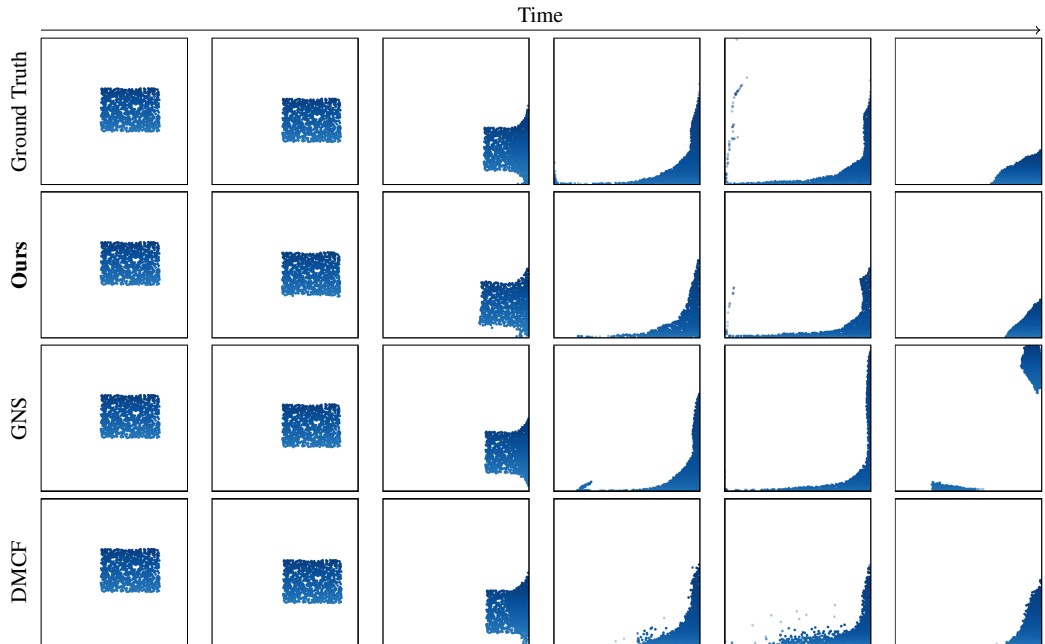

Figure 4: **Example VARIABLEGRAVITY trajectory against baselines. Each method is unrolled starting from the initial conditions of a random test trajectory not seen during training.**

Figure 3 summarizes the ablation results. A larger number $K$ of autoregressive steps yields more accurate rollouts without the need to add noise. Indeed, injecting noise does not improve accuracy and is even detrimental for $K = 4$. Individually tuning the noise levels for grids and particles can modestly lower error rates, but is either very sensitive or negligible. The model performs better when bundling more timesteps, enabling faster rollouts as a single forward pass predicts more steps. We found $m = 8$ to be optimal with the other default hyperparameters, outperforming larger bundling. This is because more network capacity is needed to extract information for the next 16 or 32 timesteps from a single state. Instead, we opted for a shallower and narrower network to balance speed and memory footprint with performance gains. In terms of network architecture, we chose a U-Net. We experiment with an FNO (Li et al., 2021) in Appendix B and find it to underperform, leading us to keep the U-Net architecture. We find the U-Net's width and depth to have a minor impact on performance, confirming that a larger network is not needed. The grid size, however, is critical. A low resolution loses fine details, while a high resolution turns meaningful structures, such as liquid blobs or walls, into isolated voxels.

## 4.2 COMPARISON WITH PREVIOUS WORK

We compare NeuralMPM against GNS and DMCF. We use the official implementations and training instructions to assess training times, inference times, as well as accuracy. We compare against both GNS and DMCF on WATERRAMPS, SANDRAMPS, GOOP, DAM BREAK 2D, and VARIABEL-GRAVITY. We also compare against GNS on MULTIMATERIAL, but not against DMCF since it does not support multiple materials.

**Accuracy.** We report quantitative results comparing the long-term accuracy in Table 1 and show trajectories of NeuralMPM in Figure 2, as well as comparisons against baselines on WATERRAMPS in Figure 4. On the mono-material datasets WATERRAMPS, SANDRAMPS, and GOOP, NeuralMPM performs competitively with GNS and better than DMCF in terms of mean squared error (MSE). For MULTIMATERIAL, NeuralMPM reduces the MSE by almost half, which we attribute to it being a hybrid method, known to better handle interactions, mixing, and collisions between different materials. In DAM BREAK 2D, NeuralMPM outperforms both baselines, despite the data being simulated using SPH. Finally, NeuralMPM surpasses the performance of DMCF in VARIABLEGRAVITY, even

| Data (Simulator) | $N$ | $T$ | NeuralMPM | | GNS | | DMCF | |
|---|---|---|---|---|---|---|---|---|
| | | | MSE↓ | EMD↓ | MSE↓ | EMD↓ | MSE↓ | EMD↓ |
| WATERRAMPS (MPM) | 2.3k | 600 | 13.92 | **68** | **11.75** | 90 | 20.45 | 105 |
| SANDRAMPS (MPM) | 3.3k | 400 | 3.12 | **61** | **3.11** | 84 | 6.22 | 91 |
| GOOP (MPM) | 1.9k | 400 | **2.18** | **55** | 2.4 | 73 | 5.25 | 85 |
| MULTIMATERIAL (MPM) | 2k | 1000 | **9.6** | **66** | 14.79 | 105 | - | - |
| DAM BREAK 2D (SPH) | 5k | 401 | **29.07** | **348** | 87.04 | 384 | 74.77 | 381 |
| VARIABLEGRAVITY (MPM) | 600 | 1000 | **14.48** | **92** | 134 | 350 | 28.77 | 97 |

Table 1: Full rollout MSE & EMD (both $\times 10^{-3}$) for NeuralMPM and the baselines on each dataset, with the maximum number of particles $N$ and sequence length $T$. Each method was trained until full convergence (NeuralMPM: 15h, GNS: 240h, DMCF: 120h), and the best model was used.

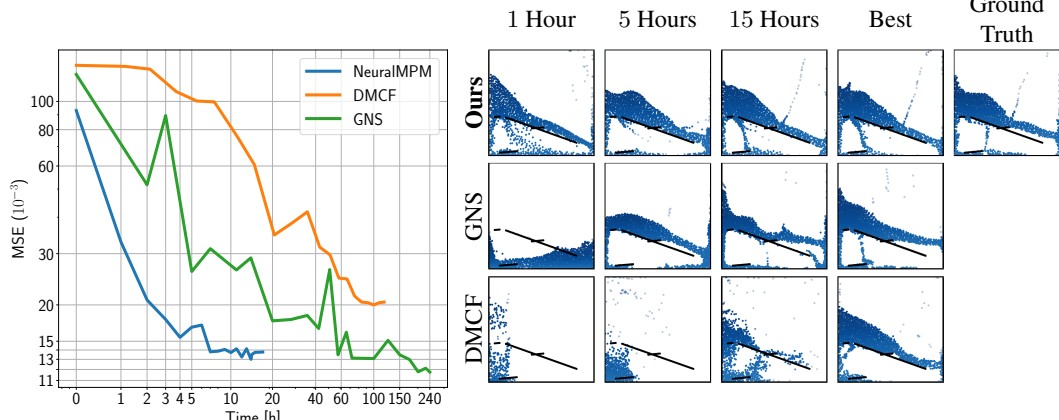

Figure 5: **Training convergence.** (Left) NeuralMPM trains and converges much faster than GNS and DMCF. Note the log scale on both axes. (Right) Snapshots of models trained for increasing durations then unrolled until the same timestep on a held-out simulation. For a fair comparison, out-of-bounds particles in GNS and DMCF were clamped.

though the latter accounts for gravity explicitly. In terms of Earth Mover's Distance (EMD), NeuralMPM outperforms both baselines across all benchmarks, suggesting that NeuralMPM is better at capturing the spatial distribution of the particles.

**Training.** In Figure 5, we report the evolution of the mean squared error of full emulated rollouts on the held-out test set during training, for each method, along with predicted snapshots at increasing training durations. NeuralMPM converges significantly faster than both baselines while reaching lower error rates. Furthermore, the convergence of the training procedure and quality of the architecture can be assessed much earlier during training, effectively saving compute and enabling the development of more refined final models. Moreover, NeuralMPM is also more memory-efficient, which enables the use of higher batch sizes of 128, as opposed to only 2 in GNS and DMCF.

**Inference time and memory.** In Figure 6, we display the time and memory performance of NeuralMPM, the two baselines GNS and DMCF, and the reference solver Taichi-MPM. In terms of speed, NeuralMPM strongly outperforms all three methods, partly thanks to time bundling, which considerably reduces the number of model calls required for a given number of frames to emulate. In terms of memory, although NeuralMPM remains inferior to Taichi-MPM, which is highly optimized, it can emulate tens of millions of particles on a single GPU, while GNS and DMCF struggle to reach half a million.

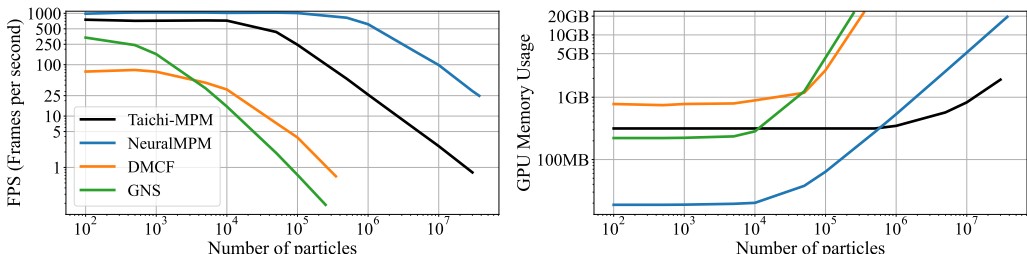

Figure 6: **Time and memory performance.** Average FPS (left) and GPU VRAM usage (right) for increasing numbers of particles for a traditional solver (Taichi-MPM (Hu et al., 2018a)), NeuralMPM, and the two baselines. The two baselines quickly require very large amounts of memory and become very slow. Although Taichi-MPM is more memory efficient for high numbers of particles, NeuralMPM remains much faster, emulating 30 million particles at 25FPS. For the low particle count regime (< 10K) we used the NeuralMPM and baselines WaterRamps models. For the high particle count regime we used untrained models and measured the throughput. The figures measures just FPS, and not the real simulation time. Taichi-MPM needs a much smaller step size than the three neural emulators ($2 \times 10^{-4}$s vs $2.5 \times 10^{-3}$s), and is therefore likely slower than all of them

## 4.3 GENERALIZATION

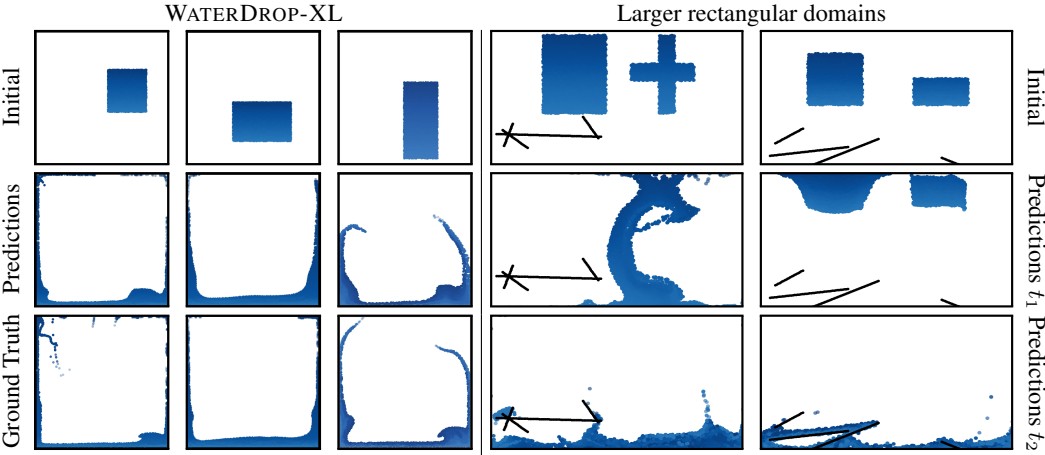

Figure 7: **Generalization.** (Left) NeuralMPM generalizes to domains with more particles ($\sim 4\times$ here) with minimal inference time overhead due to the processing of the voxelized representation. (Right) A NeuralMPM model trained on a square domain can naturally generalize to larger rectangular domains (twice as wide here) when using a fully convolutional U-Net.

One notable advantage of NeuralMPM is that the processor is invariant to the number of particles, as the transition model only processes the voxelized representation, while both **p2g** and **g2p** scale linearly. To demonstrate this, we train a model on WATERRAMPS, which contains about 2.3k particles and 600 timesteps, and evaluate it on WATERDROP-XL, which features about four times more particles, 1000 timesteps, and no obstacles. An example snapshot is displayed in Figure 7. The larger number of particles only affects interpolation steps between the grid and particles, resulting in a negligible impact on total inference time, making the model nearly as fast despite 4 times more particles. We also validate generalization quantitatively by comparing the error rates on WATERDROP-XL of a model trained directly on it and the model trained solely on WATERRAMPS. With the same training budget, the latter achieves a lower MSE at $20.92 \times 10^{-3}$ against $28.09 \times 10^{-3}$. More trajectories are displayed in Figure 22.

If a domain-agnostic processor architecture is used, such as a fully convolutional U-Net or an FNO, then NeuralMPM can generalize to different domain shapes without retraining, as shown in Fig 7. We demonstrate this ability by considering a model solely trained on WATERRAMPS, a square domain of size $0.84 \times 0.84$ mapped to $64 \times 64$ grids. Without retraining, we perform inference with this model on larger unseen environments of size $1.68 \times 0.84$, and change the grid size to $128 \times 64$. The unseen environments were built by merging and modifying initial conditions of held-out test trajectories from WATERRAMPS. NeuralMPM emulates particles in this larger and rectangular domain despite being trained on a smaller square domain with a smaller grid, showing that a U-Net can generalize to other domains. No ground truth is displayed as Sanchez-Gonzalez et al. (2020) provide no information about the data generation. More trajectories are shown in Figure 23.

### 4.4 INVERSE DESIGN PROBLEM

Finally, we demonstrate the application of NeuralMPM for inverse problems on a toy inverse design task that consists in optimizing the direction of a ramp to make the particles reach a target location, similar to (Allen et al., 2022). We place a blob of water at different starting locations, and we then place a ramp at some location, with a random initial angle $\alpha$. The goal is to spin the ramp by tuning $\alpha$ in order to make the water end up at a desired location. The main challenges of this task are the long-range time horizon of the goal and the presence of nonlinear physical dynamics. We proceed by selecting the point where we want the water to end up and compute the average distance between the point and particles at the last simulation frame. We then minimize the distance via gradient descent, leveraging the differentiability of NeuralMPM to solve this inverse design problem. We show am example optimization in Figure 8, and additional examples in Appendix B.

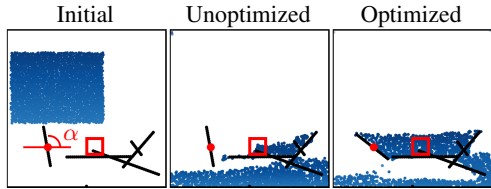

Figure 8: **Inverse design problem.** We exploit NeuralMPM's differentiability to optimize the angle $\alpha$ of a ramp, anchored at the red dot, in order to get the water close to the red square region.

## 5 CONCLUSION

**Summary.** We presented NeuralMPM, a neural emulation framework for particle-based simulations inspired by the hybrid Eulerian-Lagrangian Material Point Method. We have shown its effectiveness in simulating a variety of materials and interactions, its ability to generalize to larger systems and its use in inverse problems. Crucially, NeuralMPM trains in 6% of the time it takes to train GNS and DMCF to comparable accuracy, and is 5x-10x faster at inference time. By interpolating particles onto a fixed-size grid, global information is distilled into a voxelized representation that is easier to learn and process with powerful image-to-image models. The use of voxelization allows NeuralMPM to bypass expensive graph constructions, and the interpolation leads to easier generalization to a larger number of particles and constant runtime. The lack of expensive graph construction and message passing also allows for more autoregressive steps and parallel rollouts.

**Limitations.** Like other approaches, NeuralMPM is limited by the computation used to process the structure of the point cloud. In our case, voxelization means we cannot deal with particles that lie outside of the domain and are limited to regular grids. Additionally, the size of the voxels is directly related to the number of particles within a given volume. If the voxels are too large, the model will fail to capture finer details. Conversely, if they are too small, the model may struggle due to insufficient local structure. Similarly, performance can degrade in very sparse domains. Compressible fluids might also present challenges, though this requires further verification.

**Future work.** Our work is only a first step towards hybrid Eulerian-Lagrangian neural emulators, leaving many avenues for future research. Extending NeuralMPM to 3D systems is a natural continuation of this work. Future studies could also explore alternative particle-to-grid and grid-to-particle

functions, like the non-uniform Fourier transform (Fessler & Sutton, 2003), or more sophisticated interpolation methods from classical MPM literature (Nguyen et al., 2023). A less traditional direction is to make NeuralMPM probabilistic and encode richer distributional information about the particles in the grid nodes, instead of maintaining only a mean value. This could potentially improve NeuralMPM's ability to resolve subgrid phenomena. Finally, advances in Lagrangian Particle Tracking (Schröder & Schanz, 2023) will eventually make it possible to create datasets from real-world data, enabling the training of NeuralMPM directly from data without the need for the costly design process of a numerical simulator.

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

## A    TRAINING DETAILS

**Hardware.**    We run all our experiments using the same hardware: 4 CPUs, 24GB of RAM, and an NVIDIA RTX A5000 GPU with 24GB of VRAM. For reproducing the results of DMCF, we kept the A5000 GPU but it required up to 96GB of RAM for training.

**Data Preprocessing.**    Similar to Prantl et al. (2022), we slightly alter the original MPM datasets to add boundary particles, as the original data from Sanchez-Gonzalez et al. (2020) does not have them. We define the velocity at a timestep to be $\mathbf{v}_t = \mathbf{v}_t - \mathbf{v}_{t-1}$. We therefore skip the first step during training for which no velocity is available.

**Implementation.**    Our implementation, training scripts, experiment configurations, and instructions for reproducing results are publicly available at [URL]. We implement NeuralMPM using PyTorch (Paszke et al., 2019), and use PyTorch Geometric (Fey & Lenssen, 2019) for implementing efficient particle-to-grid functions, more specifically from the Scatter and Cluster modules. For memory efficiency, we do not store all (up to) 1,000 training trajectories in memory, and rather use a buffer of about 16 trajectories over which several epochs are performed before loading a new buffer of random trajectories.

**Baselines.**    We use the official implementations and training instructions of GNS (Sanchez-Gonzalez et al., 2020) and DMCF (Prantl et al., 2022) to reproduce their results and conduct new experiments. More specifically, we train GNS as instructed for 20M steps on all four datasets, using their provided configuration. For DMCF, we follow their default configurations and train for 400K iterations for each dataset. In datasets that were not used by the original authors, VARIABLEGRAVITY and DAM BREAK 2D, we performed hyperparameter search. GNS and DMCF both were trained for a total budget of 60 GPU-days per dataset, and the best performance was reported.

**Normalization.**    We normalize the input of the model over each channel individually. We investigated computing the statistics across a buffer, resembling (Ioffe & Szegedy, 2015), and precomputing them on the whole training set and found no difference in performance. During inference, we use the precomputed statistics.

**Code.**    The code, together with additional videos, is available at the project's website [URL].

# B  SUPPLEMENTARY RESULTS

**Additional results on DamBreak2D**  We have also compared the accuracy and inference speed of NeuralMPM against a different implementation of GNS, and one of SEGNN, both provided by (Toshev et al., 2024), in Tables 2 and 3. As in Table 1, NeuralMPM outperforms both by a margin baselines.

|        | MSE↓   | EMD↓  |
|--------|--------|-------|
| **Ours** | **20.76** | **2.88** |
| GNS    | 114.40 | 224.1 |
| SEGNN  | 124.39 | 268.4 |

Table 2: Full-trajectory MSE ($\times 10^{-3}$) and Sinkhorn distance (EMD) ($\times 10^{-4}$) for NeuralMPM, GNS, and SEGNN Brandstetter et al. (2022a) on the DAM BREAK 2D dataset from LagrangeBench. The two latter models are baselines provided by LagrangeBench.

|        | Single call ($T = 1$) | Rollout ($T = 401$) |
|--------|-----------------------|---------------------|
| **Ours** | **7.41**            | **193.50**          |
| GNS    | 20.46                 | 8,170.47            |
| SEGNN  | 46.04                 | 18,194.10           |

Table 3: Inference time (in ms) of NeuralMPM, GNS, and SEGNN Brandstetter et al. (2022a) on the DAM BREAK 2D dataset from LagrangeBench. Times were averaged over all test trajectories. NeuralMPM predicts 16 frames in a single model call and still outperforms the two baselines per call, which further widens the gap for the total rollout time.

**Evaluation.**  In Table 4, we report the numerical MSE rollout values that were reported in the bar plots depicted in Figure 3 for GOOP. Also, Figures 11 and 10 displays the error when rolling out a model for each dataset, both in terms of MSE and EMD. For both metrics, the error starts low and slowly accumulates over time. For the EMD, we use the Sinkhorn algorithm provided by (Cuturi et al., 2022).

| Parameter | Value | MSE ($\times 10^{-3}$) | Parameter | Value | MSE ($\times 10^{-3}$) |
|-----------|-------|------------------------|-----------|-------|------------------------|
| $K$ (No noise) | 1 | 3.2 | Grid noise | 0 | 3.2 |
|           | 2 | 3.3 |            | 0.001 | 2.4 |
|           | 3 | 2.4 |            | 0.005 | 6.9 |
|           | 4 | 2.2 | Particle noise | 0 | 2.2 |
| $K$ (With noise) | 1 | 3.5 |        | 0.0003 | 2.4 |
|           | 2 | 2.5 |            | 0.0006 | 2.4 |
|           | 3 | 2.4 |            | 0.001 | 2.1 |
|           | 4 | 3.0 | U-Net Depth | 2 | 3.3 |
| Time bundling $m$ | 1 | 6.6 |          | 3 | 3.0 |
|           | 2 | 4.5 |            | 4 | 2.4 |
|           | 4 | 3.5 |            | 5 | 2.3 |
|           | 8 | 2.1 | U-Net Width | 32 | 2.6 |
|           | 16 | 2.9 |           | 64 | 2.3 |
|           | 32 | 3.5 |           | 128 | 2.2 |
| Grid size | 32 | 5.5 | | | |
|           | 64 | 2.4 | | | |
|           | 128 | 7.1 | | | |

Table 4: Ablation results for GOOP.

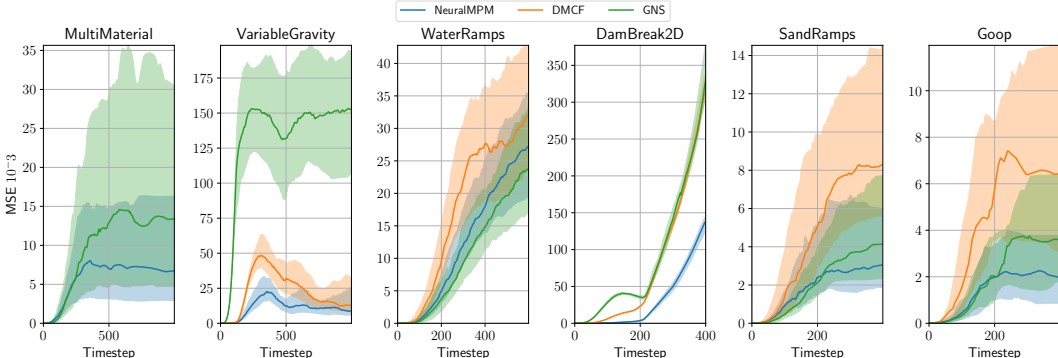

Figure 9: **MSE propagation during rollout.** We show the 25th, 50th and 75th percentile of the MSE, computed over particles and simulations, at each timestep during the rollout for each model and dataset. The accuracy decreases as errors accumulate. While training in 5% of the time, NeuralMPM outperform the baselines on all datasets, except WaterRamps, where it is slightly worse than GNS.

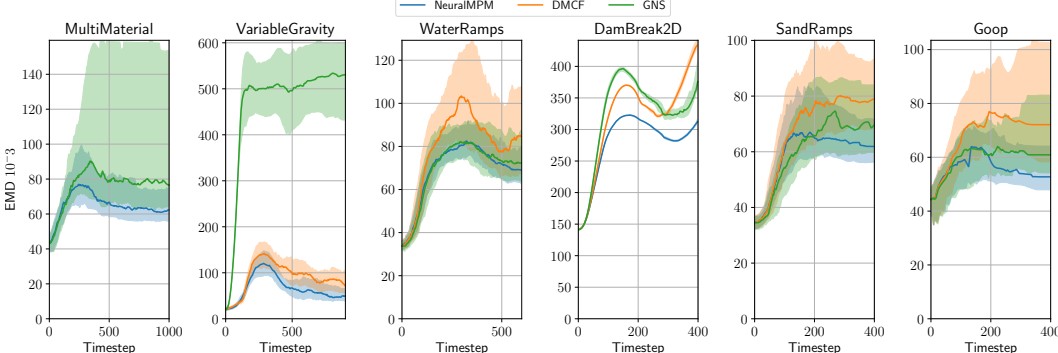

Figure 10: **EMD propagation during rollout.** We show the 25th, 50th and 75th percentile of the EMD, computed over particles and simulations, at each timestep during the rollout for each model dataset daset. The accuracy decreases as errors accumulate. While training in 5% of the time, NeuralMPM outperform the baselines on all datasets.

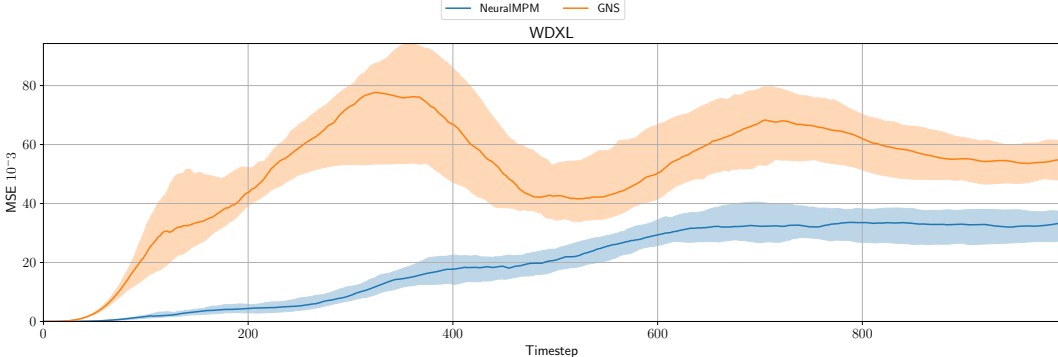

Figure 11: **MSE propagation during rollout for the generalization task WaterDrop-XL.** We show the 25th, 50th and 75th percentile of the MSE, computed over particles and simulations, at each timestep during the rollout for each model and daset. The accuracy decreases as errors accumulate. The models used were trained on WaterRamps and tested on WaterDrop-XL to evaluate their generalization. NeuralMPM performs better despite having a slightly worse MSE on WaterRamps than GNS.

**Grid-to-grid network.** Although we have used a U-Net architecture for the grid-to-grid processor, NeuralMPM can be used with any grid-to-grid processor and is not limited to that network. For example, in Figure 12 and Table 5 we present qualitative and quantitative ablation results, respectively, for NeuralMPM using an FNO network (Li et al., 2021) as the grid-to-grid processor. Results show that the FNO processor is slightly worse than the U-Net processor.

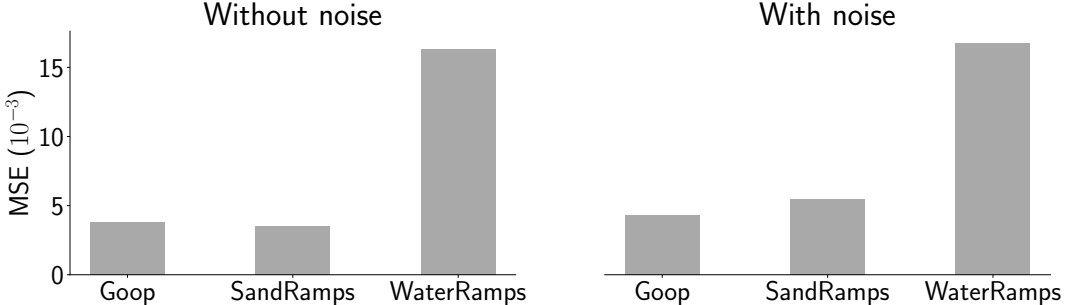

Figure 12: **FNO processor.** NeuralMPM with an FNO processor and default architecture. Rollout MSE ($\times 10^{-3}$) for different datasets.

| Data | FNO with noise | FNO without noise |
|---|---|---|
| WATERRAMPS | 16.8 | 16.3 |
| SANDRAMPS | 5.5 | 3.5 |
| GOOP | 4.3 | 3.8 |

Table 5: Rollout MSE ($\times 10^{-3}$) for NeuralMPM with an FNO processor and default architecture, with and without noise.

**Additional inverse problem examples.** We show two additional optimization examples in Figure 13.

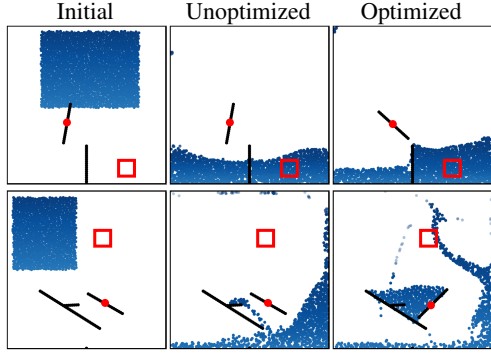

Figure 13: **Inverse design problem.** Additional optimization examples. We exploit NeuralMPM's differentiability to optimize the angle $\alpha$ of a ramp, anchored at the red dot, in order to get the water close to the red square region.

# C  GALLERY OF PREDICTED TRAJECTORIES

We present additional rollout comparisons in Figures 14 and 15. Further, in addition to the trajectories in Figures 2 and 4, we show additional trajectories emulated with NeuralMPM for all datasets in Figures 16, 17, 18, 19, 20, 21, 22, and 23. We also release *videos* in the supplementary material, which we recommend watching to better see the details and limitations of NeuralMPM. This includes 10 videos per dataset of emulated trajectories on held-out test simulations.

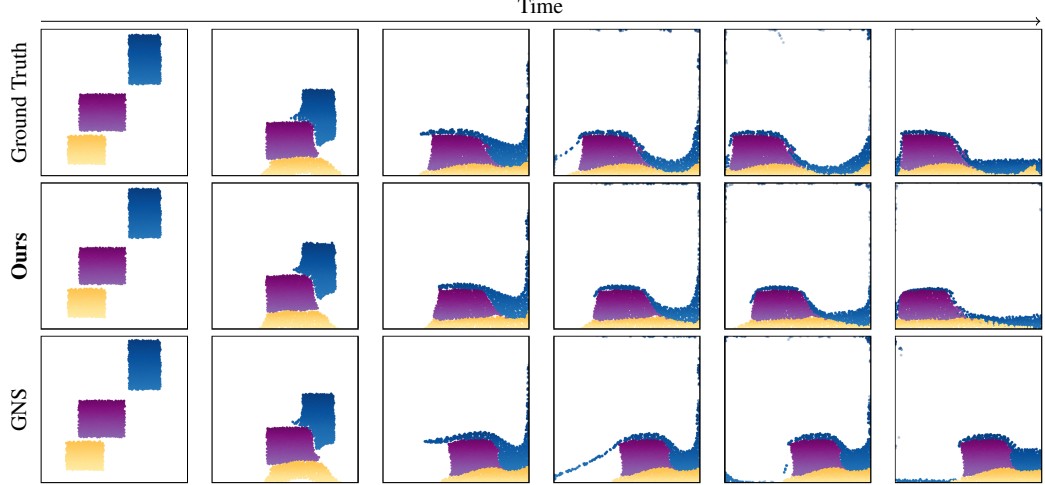

Figure 14: **Example MULTIMATERIAL trajectory against baselines. Each method is unrolled starting from the initial conditions of a random test trajectory not seen during training.**

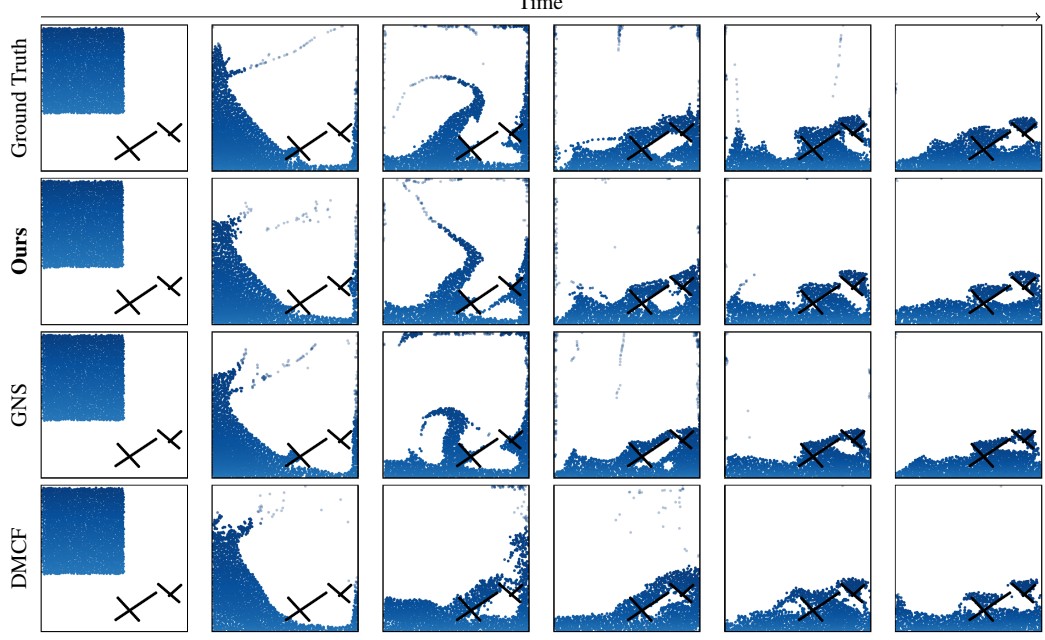

Figure 15: **Example WATERRAMPS trajectory against baselines. Each method is unrolled starting from the initial conditions of a random test trajectory not seen during training.**

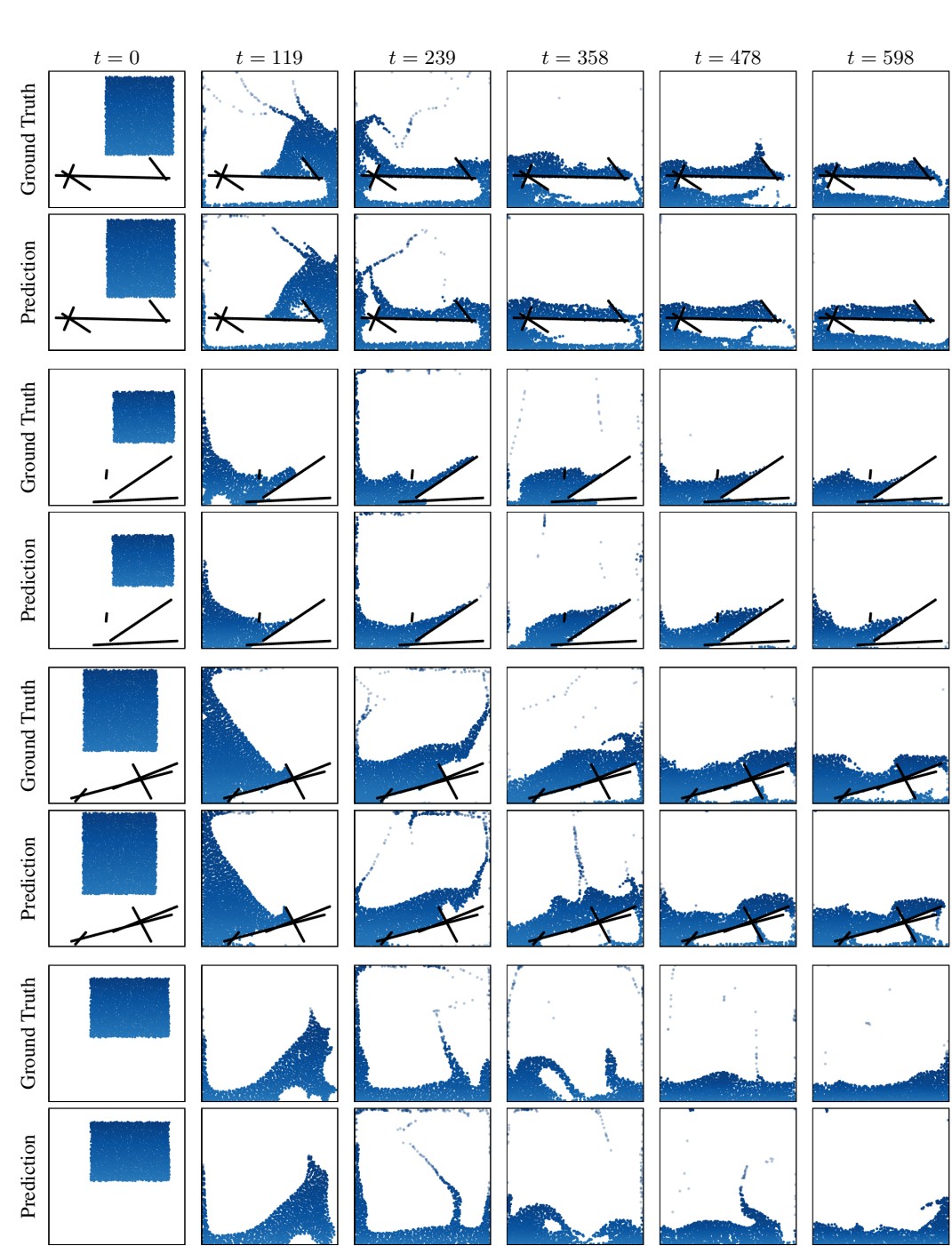

Figure 16: **Additional WATERRAMPS predicted trajectories.** Evenly spaced in time snapshots of predicted unrolled trajectories against ground truth. All trajectories are from the held-out test set.

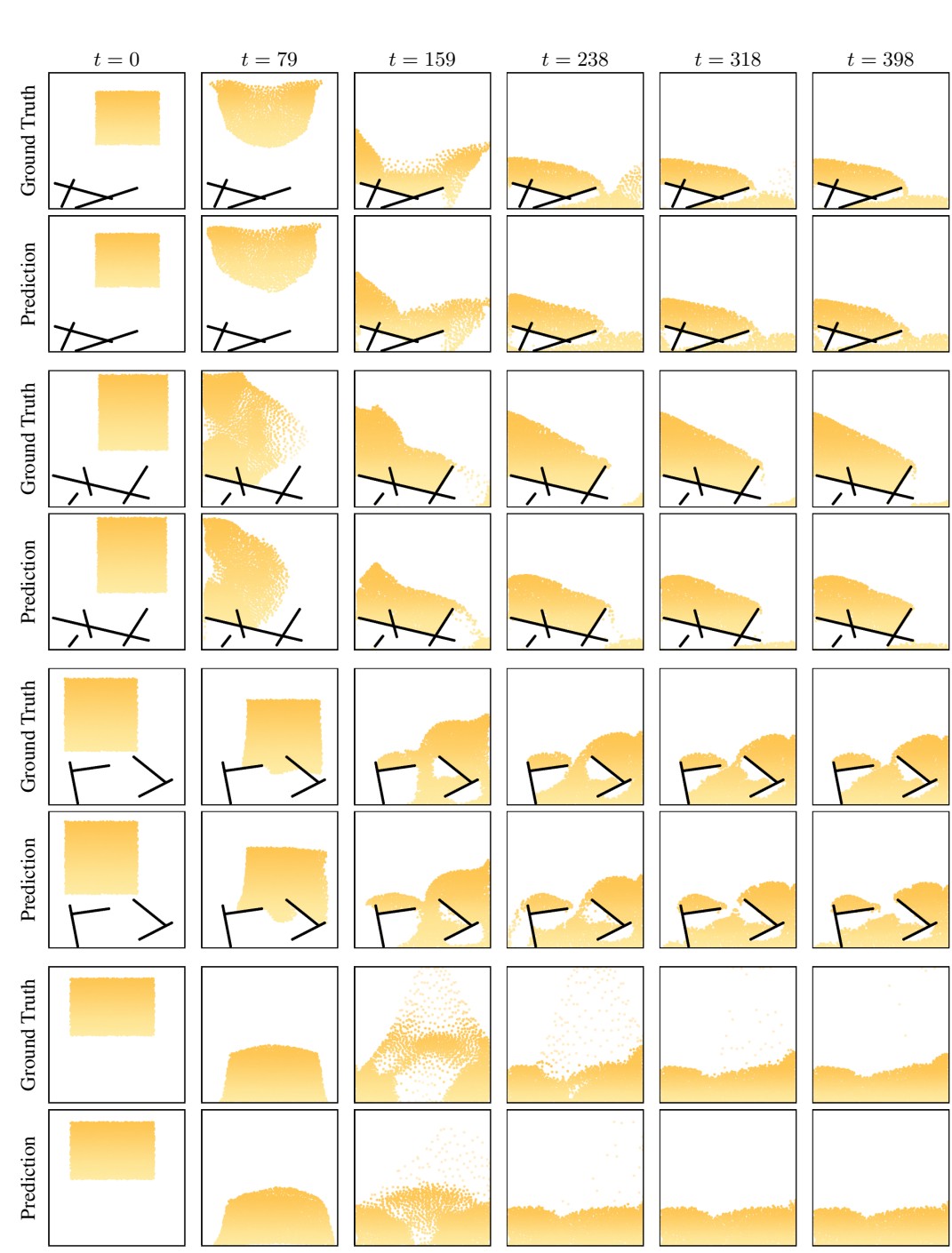

Figure 17: **Additional SANDRAMPS predicted trajectories.** Evenly spaced in time snapshots of predicted unrolled trajectories against ground truth. All trajectories are from the held-out test set.

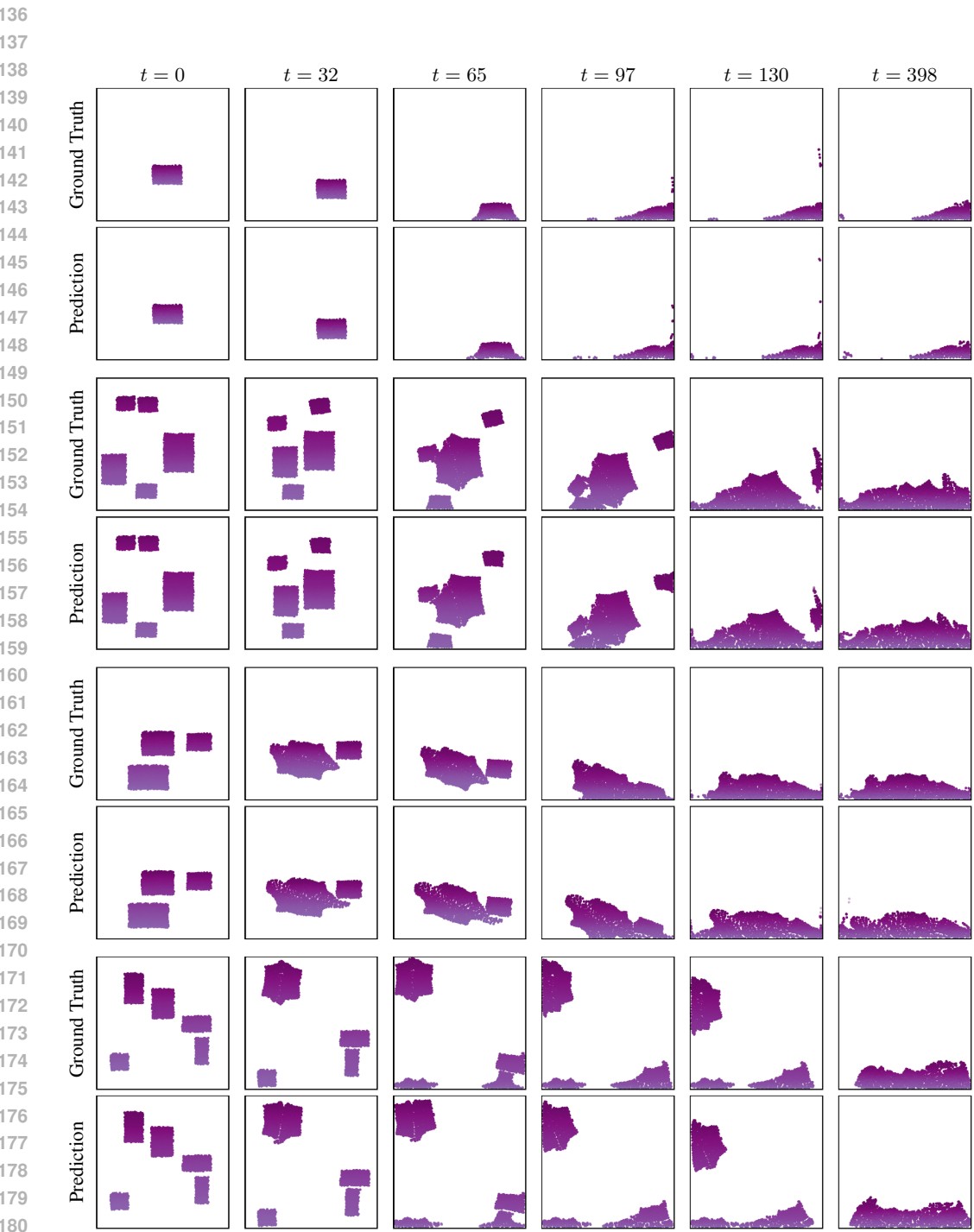

Figure 18: **Additional GOOP predicted trajectories.** Snapshots of predicted unrolled trajectories against ground truth. All trajectories are from the held-out test set. Due to GOOP quickly reaching equilibrium, more snapshots are taken in the first half of the trajectory.

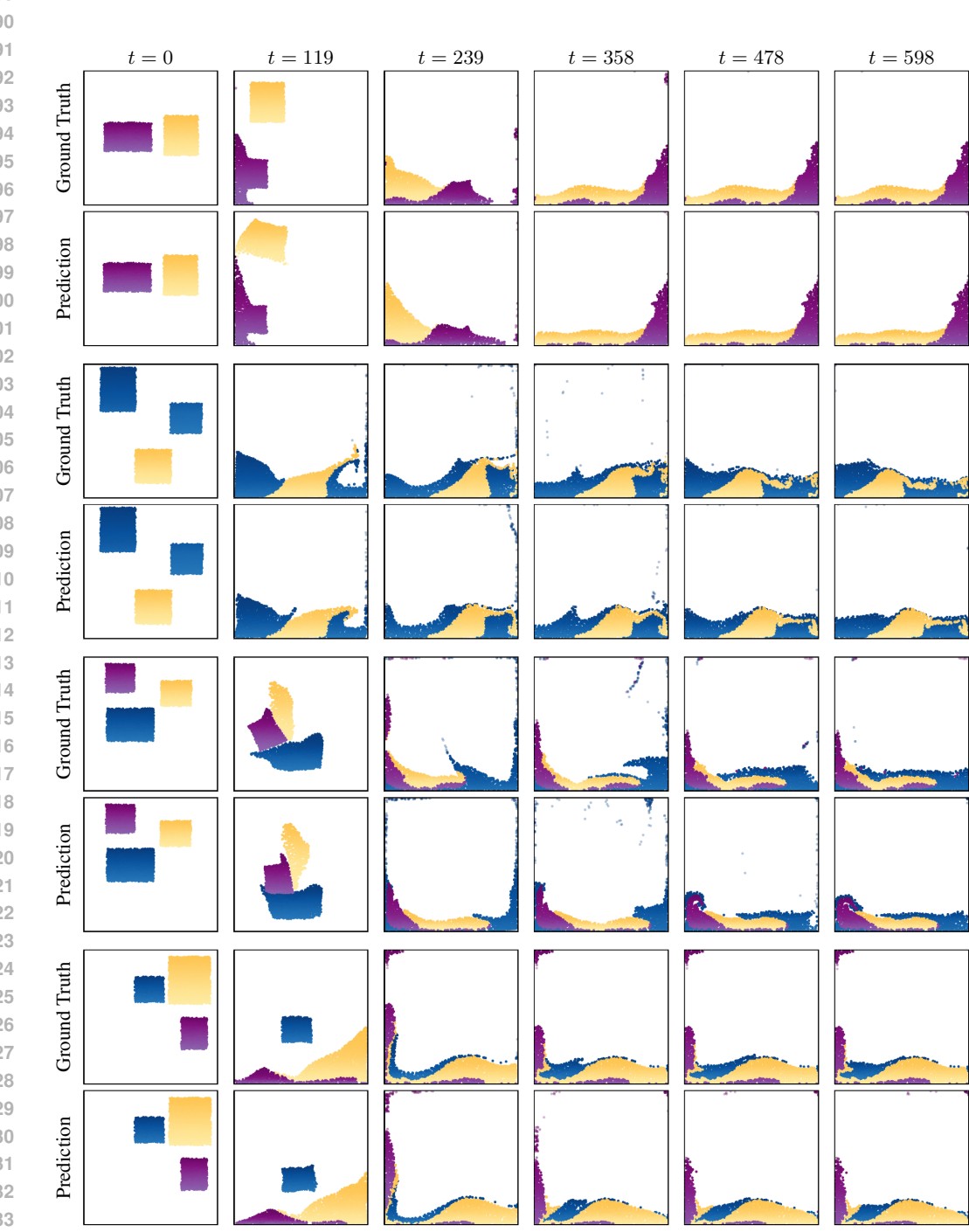

Figure 19: **Additional MULTIMATERIAL predicted trajectories.** Evenly spaced in time snapshots of predicted unrolled trajectories against ground truth. All trajectories are from the held-out test set. The first trajectory illustrates a rare failure where the shape of sand particles is not retained, even though all particles are supposed to maintain the same velocity while airborne, as they are thrown against the wall.

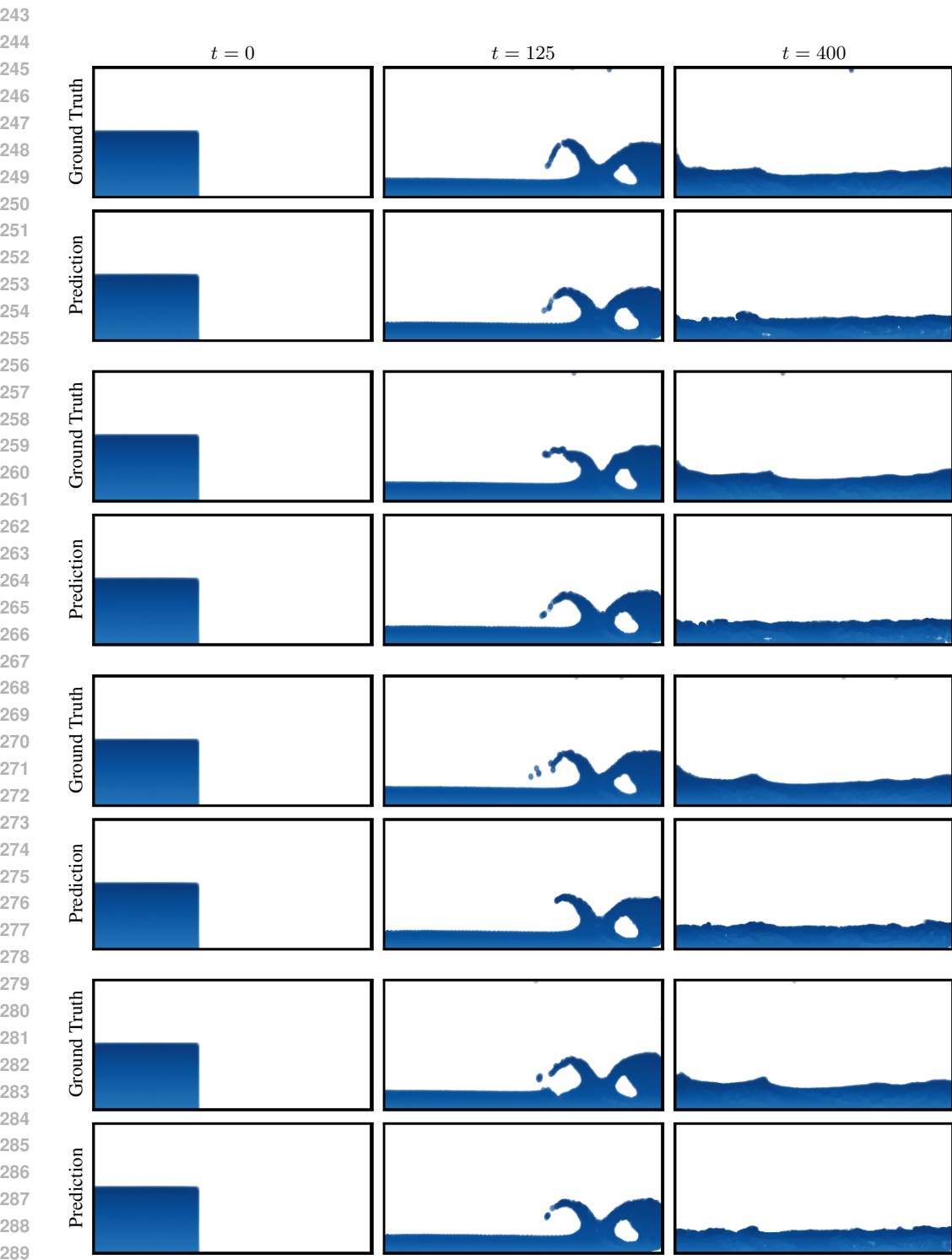

Figure 20: **Additional DAM BREAK 2D predicted trajectories.** Snapshots of predicted trajectories against ground truth. All trajectories come from the held-out test set. To better show the differences of these longer sequences, we select the following timesteps not even in time: $t \in \{0, 125, 400\}$. In the last trajectory, NeuralMPM struggles to follow the gravity direction and breaks down over time.

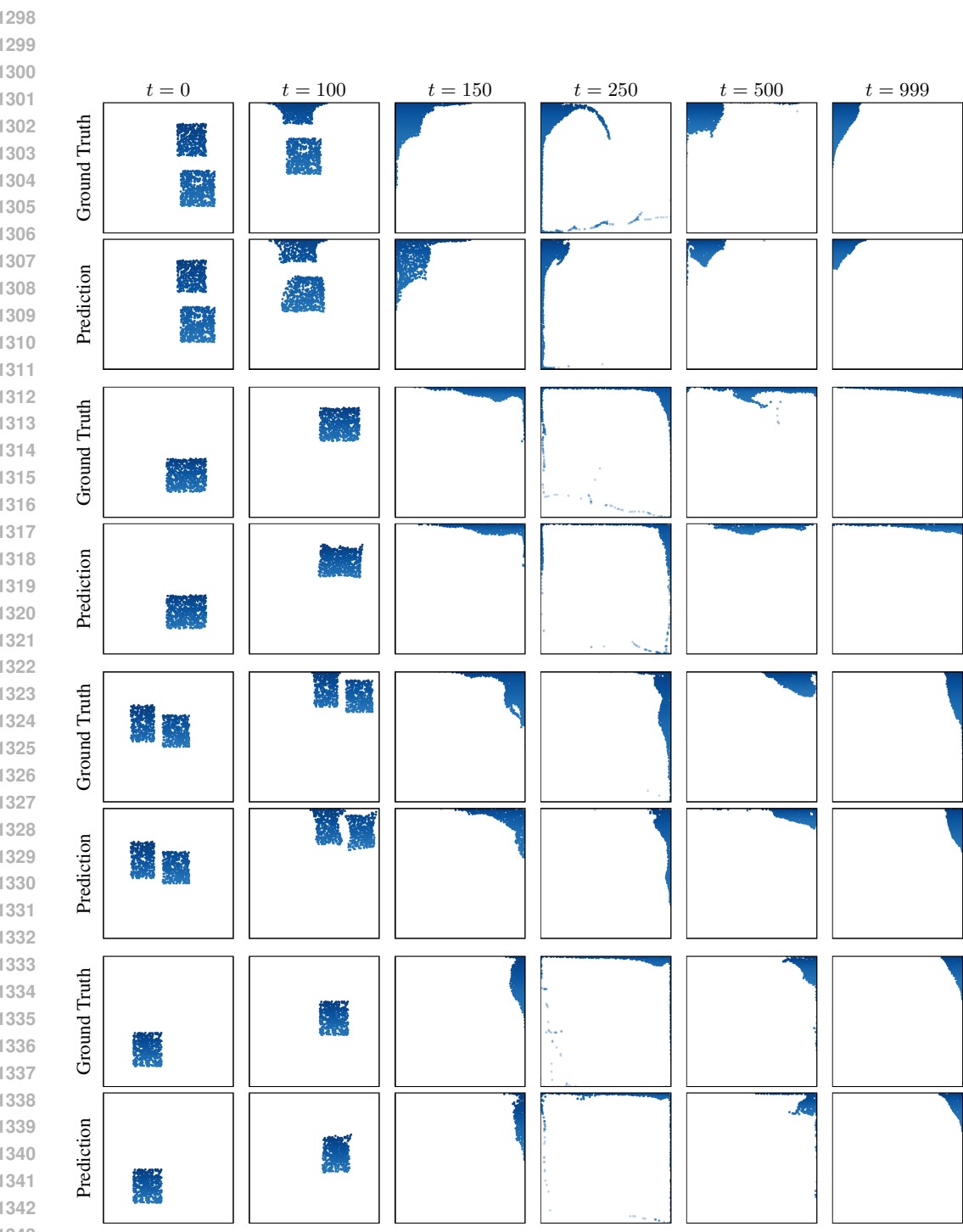

Figure 21: **Additional VARIABLYGRAVITY predicted trajectories.** Snapshots of predicted trajectories against ground truth. All trajectories come from the held-out test set.

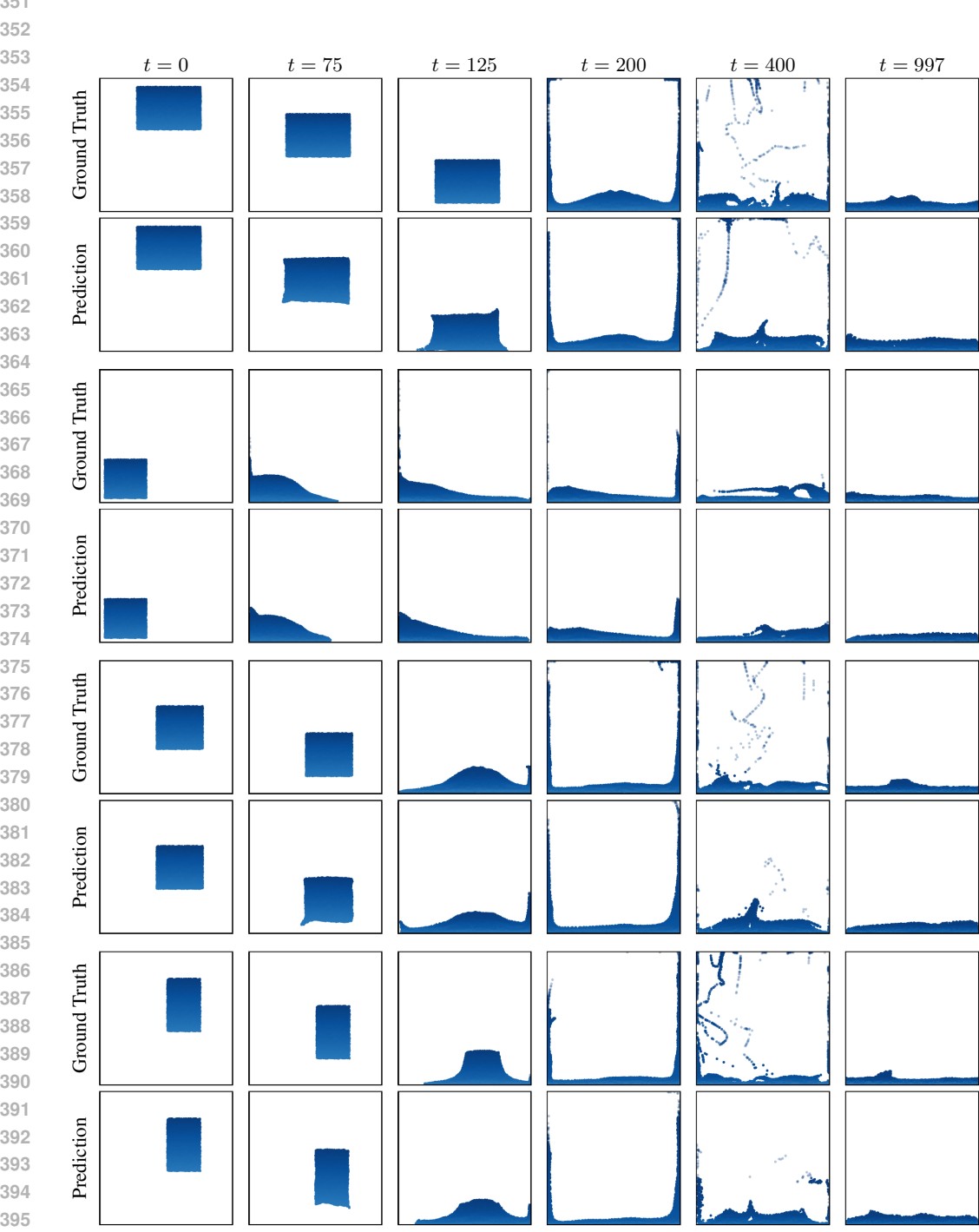

Figure 22: **Generalization to more particles on WATERDROP-XL.** Snapshots of predicted trajectories emulated using a model trained solely on WATERRAMPS, against ground truth. All trajectories come from the held-out test set of WATERDROP-XL. To better show the differences of these longer sequences, we select the following timesteps not even in time: $t \in \{0, 75, 125, 200, 400, 999\}$. We can observe that the generalizing model struggles to retain the shape of water while it's falling.

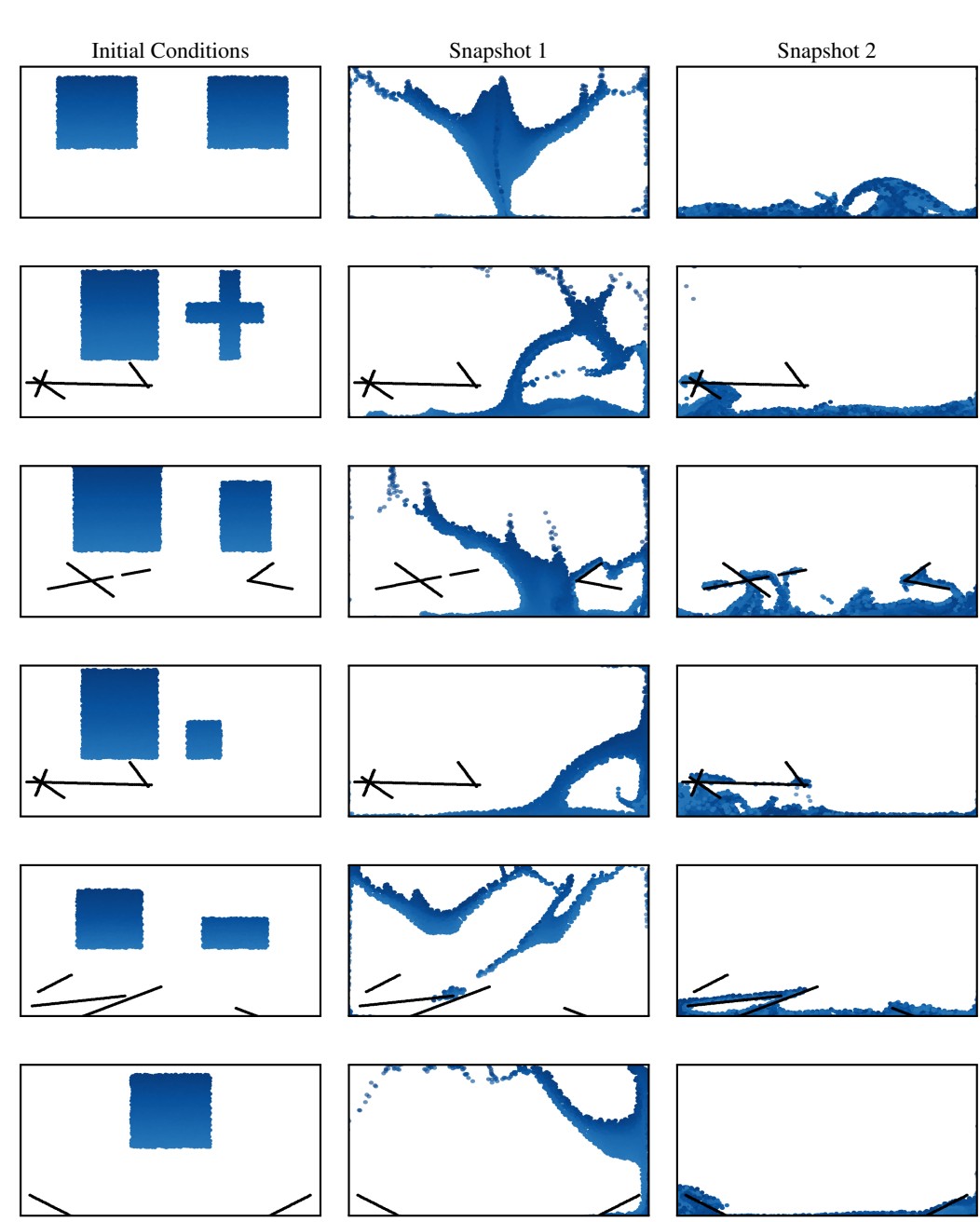

Figure 23: **Generalization to larger and non-square domains.** We train a model on the square domains in WATERRAMPS using $64 \times 64$ input grids to the U-Net, and then perform inference on manually generated non-square environments that are twice as wide and use a $128 \times 64$ input grid to the same U-Net. NeuralMPM flawlessly generalizes and emulates particles in these new environments. Note: no ground truth is available because the authors of GNS did not provide the physical parameters for simulating WATERRAMPS using Taichi. Chosen time steps are $0, 150, 575$. We recommend watching the videos in the supplementary material for more detailed evaluation.

