# OpenReview forum: "A Neural Material Point Method for Particle-based Simulations"
_ICLR.cc/2025/Conference — ICLR 2025 Conference Withdrawn Submission_

### Official Review · Reviewer_DPie · 2024-10-17

**Soundness:** 2
**Presentation:** 2
**Contribution:** 2
**Rating:** 6
**Confidence:** 3

**Summary:**

The paper presents NeuralMPM, a learning-based PDE solver inspired by the Material Point Method. It falls in the category of Lagrangian (or particle-based) methods, but the MPM, by interpolating Lagrangian particles onto a fixed-size grid via voxelization, makes it a hybrid Eulerian-Lagrangian approach.

The advantage of NeuralMPM is the fact that it still carries information via particles, i.e. it is not mesh-based, like Eulerian methods, but it borrows from such methods the regular grid structure to compute the state dynamics, which is computationally less expensive than particle-based approaches. The voxelization onto a grid makes NeuralMPM amenable to networks that work in a Euclidean setting.

NeuralMPM is validated on 6 datasets, including fluid dynamics and fluid-solid interactions, with different materials and physical properties. NeuralMPM outperforms 2 other architectures, GNS and DMCF, in 4 out of 6 scenarios, converges quickly (10-fold improvement) and its performances in terms of speed and computation are more robust as the number of particles of the system increases.

**Strengths:**

The text is clearly written. It reads well, the introduction on CFD methods and the review of the MPM approach is clear and thorough, the problem setting of the NeuralMPM is clearly defined. Ablations are thorough and well presented. I liked how conclusions were drawn. The limitations of NeuralMPM and possible future work showed good confidence in this field of research.

I feel the main strengths are Fig. 5, relative to convergence, and Fig. 6 relative to the model’s robustness show important contribution of NeuralMPM, as computational complexity is one of the main limitations of learning-based PDEs solvers.

**Weaknesses:**

Reading Section 2, I am not sure I quite understand how novel is NeuralMPM. How does it compare to other methods that use MPM in a learning setting? How does it differ from the work of Li et al. (2024)? What are its main contributions?

NeuralMPM is compared against only two models, namely GNS and DMCF, making it hard to assess how good it actually is.

In general, I don’t think authors make a compelling argument for the merits of their model, probably due to their choice of figures. A list below:

 1. why using the WaterRamps dataset in Figure 4 if WaterRamps is the only dataset over which NeuralMPM does not outperform the other two models?

2. Figure 7: you claim that ”A NeuralMPM model trained on a square domain can naturally generalise to larger rectangular domains (twice as wide here) when using a fully convolutional U-Net”, but without ground truth (as you claim in Fig. 19 and line 486) for the right hand side particles how can we assess how good is NeuralMPM at generalising? I am having a hard time believing that the evolution from starting point to $t_1$ is realistic (where did the leftmost blob of water go?).
Also, the labels between the left and right section of the figure are confusing (Predictions and GT for left and Predictions $t_1$, Predictions $t_2$ for right).

3. In line 469 you claim that “one notable advantage of NeuralMPM is its invariance to the number of particles, as the transition model only processes the voxelized representation.” Doesn’t figure 6 confute the claim? After 10^5 particles the FPS decreases and after 10^4 the GPU usage increases for NeuralMPM. They don’t seem to be an incredibly large number of particles either.  Could you please clarify this point?


4. You claim that NeuralMPM is superior to GNS, but the results reported by their authors show otherwise, as shown in Fig. 3. Without a discussion on what might cause this discrepancy, it is hard to assess the merits of NeuralMPM over GNS.

The abstract and conclusions don’t highlight the merits of the paper enough, in my opinion, especially because there’s plenty of details and quantitative results in Section 4. For example:  Line 22: We demonstrate the advantages of NeuralMPM on several datasets (how many? Which ones?), including fluid dynamics and fluid-solid interactions
Line 23: Compared to existing methods (which methods?), NeuralMPM reduces training times from days to hours (can you quantify the improvement?) Line 516: NeuralMPM trains in a fraction of the time (what fraction?) it takes to train alternative approaches (what are alternative approaches?)

I am willing to increase my score if:
 1) comparisons of the models’ performance for different datasets (Multilateral, DamBreak 2D and/or Variable Gravity) are plotted, showing the 3x improvement reported in Table 1 of NeuralMPM over GNS and DMCF for those datasets also through figures.

2) an intuition on why the authors’ implementation of GNS performs differently to the original paper is provided and a comparison of the GNS training speed originally reported is offered.

3) The claim of invariance with respect to number of particle numbers is adequately defended.

4) Abstract and conclusions are reformulated to better highlight the merits and the original contributions of NeuralMPM with respect to previous literature by quantitative results.

**Questions:**

Is this the first approach in the literature of a differentiable MPM method?

Line 187: you mention that the FNO underperforms. What is your intuition behind a neural operator performing worse on a PDE solution task as opposed to a purely convolutional architecture like U-Net?

Why have you chosen specifically GNS and DMCF? It would be good to expand on that in the text.

Fig. 3: interesting that you could not reproduce the reported GNS results when you retrained it. Do you have an explanation for that? Is it due to different hyperparameters, different hardware, etc? Without further insight it’s difficult to conclude whether NeuralMPM outperforms GNS or not.

Fig. 3: Why didn’t you include a line also for the DMCF approach? Couldn’t you retrain it like you retrained GNS?

Table 1: why aren’t there results reported obtained via classical Taichi-MPM?

You report full convergence of the GNS method after 240h. What is the convergence time reported by the original authors?

In Fig. 4 you provide an example of trajectories for the WaterRamps dataset for your NeuralMPM approach and the GNS and DMCF. In Table 1, however, WaterRamps is the only dataset over which NeuralMPS shows no improvement compared to other methods. Why have you chosen to plot results over the WaterRamps dataset? Have you considered plotting examples for MultiMaterial, DamBreak or Variable Gravity? Since the improvement over the latter datasets is so significant, it would make sense to corroborate the results with figures. You have 8 additional figures in the appendix, why not having also figure of comparison on those datasets to make your argument more compelling?

Line 434: How do you explain the faster convergence of NeuralMPM? Is it due to the voxelization that makes computations easier? Also, in Figure 6 there’s the Taichi-MPM method, but that is missing in Figure 5. How does the convergence of the Taichi-MPM method compares?

**Details Of Ethics Concerns:**

no ethics concerns

---

> ### Author Response · Authors · 2024-11-15
>
> Thank you for your review, we appreciate the actionable feedback provided, and hope to have addressed the issues you raised. We will include the changes proposed below in a revision within the rebuttal period.
>
> ----
> Regarding the weaknesses:
>
> 1) Li et al.'s method is a hybrid classic-ML approach. As such it benefits from the strong inductive biases of the physics of the method, but its range of applicability is smaller. On the other hand, NeuralMPM is a fully data-driven emulator, and, as such, it can be applied to a wider range of problems at the price of less powerful inductive biases. For example, NeuralMPM is capable of accurately simulating SPH data (Table 1), a task which we believe would challenge MPMNet as it explicitly implements its physical model.
>
> 2) We chose GNS and DMCF as they are well-established, particle-based fully data-driven emulators that assume no knowledge of the dynamics. Another GNN-based method, SEGNN [1], is provided by the LagrangeBench benchmark suite that we used for Dam Break 2D. We also outperform this method and will add it in the appendix.
>
> 3) You make a good point. We would like to note that the performance on WaterRamps in terms of MSE is close, and is better in terms of EMD for NeuralMPM than for the baselines. We will update Fig. 4 to show a different dataset with a more convincing result.
>
> 4) We are sorry for the confusion. We will create the ground truth and update the figure to show the ground truth for the right hand side particles. We will also clarify the labels.
>
> 5) The processor is invariant to the number of particles, but not the whole pipeline (particle-wise operation like position updates still scale like O(N) in both memory and time). In Fig. 6 we show that the FPS remain stable longer than the baselines and the classical simulator, and that the memory consumption is orders of magnitude lower than the baselines. We will update the claim in L469 to clarify the confusion.
>
> 6) The result in Fig. 3 shows that NeuralMPM achieves a lower MSE than GNS using the implementation and hyperparameters provided by Sanchez-Gonzalez et al. [2]. We will add a discussion of the discrepancy which we attribute to the hardware they run their model on. Further, we will remove the discrepancy from Fig. 3, as we were not able to reproduce it.
>
> 7) This is a good point and we thank you for these suggestions. We will update the abstract and the conclusion to be more precise and better highlight these facts.
>
> 8) We thank you for the constructive feedback, and your clarity and willingness to increase the score. We hope to have addressed your concerns below:
>
> 9) We will add the comparison in the appendix, using the error over time metric (similar to Fig. 9), and videos for comparison.
>
> 10) We have not reimplemented GNS, but rather used the implementation by the original authors, Sanchez-Gonzalez et al. [2]. The implementation in the GNS paper is claimed to have used TPUs, which might explain the discrepancy. We ran NeuralMPM and GNS on the same hardware to ensure a fair comparison.. We will add the TPU clarification in the paper.
>
> 11) We will add a section in the appendix, and videos, where we increase the number of particles and present the results, while clarifying the claimed invariance to be limited to the processor and not the full pipeline. Note that this still results in large memory gains.
>
> 12) As stated above, we will address this point.
>
> ------
> Answers to questions: see comment below

---

> ### Author Response · Authors · 2024-11-15
>
> Answers to questions:
>
> 1) There are differentiable implementations of classic MPM methods, such as with the Diff-Taichi framework. These are not neural emulators, but rather classical simulators.
>
> 2) The input density field and output velocity field present sharp discontinuities (either there are particles, or there are not), which makes it harder to approximate the function with a Fourier-based neural operator.
>
> 3) See weakness 2)
>
>
> 4) It might be due to the difference in hardware, even though we trained longer than the original authors. It might also be due to implementation differences, as their public implementation is not the one they used on TPUs (see the bottom line of their README page on Github).
>
> 5) We originally did not think it fair, as unlike GNS DMCF did not use Goop in their paper. We have retrained DMCF on Goop (Table 1) and will include a line in Fig. 3
>
> 6) Taichi-MPM is the ground truth to which we compare against, it would simply have 0 MSE as we would be comparing a trajectory against itself.
>
> 7) They mention it only in passing: "[...]  on GPU/TPU hardware, was typically within a few hours for smaller, simpler datasets, and up to a week for the larger, more complex datasets."
>
> 8) You raise a valid point. We chose WaterRamps because it was the dataset we used to iterate during the research (more complex than Goop but simpler than the others). It is a legacy decision that we will correct by showing the results on another dataset, as stated above.
>
> 9) See answer to **sSsx Q2**. In addition, Taichi-MPM is the ground-truth classical simulator we use. While it was insightful to include it in Figure 6 for performance, it does not converge to anything as it explicitly defines the physical model.
>
> --------
> **Overall proposed changes:**
> - Add SEGNN comparison
> - Replace WaterRamps by a more meaningful dataset for Fig. 4
> - GT for rectangular domains
> - Clarify labels for rectangular domains
> - Clarify "NeuralMPM is invariant..." -> Processor is invariant and the pipeline benefits from this
> - Update abstract & conclusion with more quantitative details: how many datasets, etc.
> - More comparative plots for datasets like Multimaterial, Dam2D and VG to show the gap over GNS/DMCF.
> - Clarify the differences/discrepancy with our retraining of GNS vs. the reported results by Sanchez-Gonzalez et al.
> - Remove discrepancy in Fig. 3.
> - Clarify why we choose DMCF & GNS in the text
> - Add line for DMCF in the ablation study
>
> We thank you again and hope to have addressed your concerns and that you will reconsider your score given our answers and the proposed changes.
>
> ----
> References:
>
> [1] Brandstetter, J., Hesselink, R.D., Pol, E.V., Bekkers, E.J., & Welling, M. (2021). Geometric and Physical Quantities improve E(3) Equivariant Message Passing. ArXiv, abs/2110.02905.
>
> [2] Sanchez-Gonzalez, A., Godwin, J., Pfaff, T., Ying, R., Leskovec, J., & Battaglia, P.W. (2020). Learning to Simulate Complex Physics with Graph Networks. ArXiv, abs/2002.09405.

---

> ### Comment · Reviewer_DPie · 2024-11-26
> **Rebuttal**
>
> Sorry for the delay. I have read your rebuttal thoroughly, but some points have not been addressed thoroughly, in my opinion. I have arranged them in order of relevance.
>
> **1. On Fig. 7 and generalisability**
>
> I should have stressed this enough, but this is quite crucial to me. My concern is that the evolution of the fluid with time does not seem believable from a physical perspective. Of the two blobs (square and cross-shaped), is hard to believe that the leftmost one is completely missing at time $t_1$. Same idea goes for the right-most figure. Why does the left blob flatten on the ceiling?
>
> The authors have not commented on this. This is also shown in Fig. 19, where Snapshot 1 simply does not look like a believable evolution of the initial condition for basically all the figure shown. **The only reasonable explanation to this is that the snapshot has been taken too far in time with respect to $t_0$**.
>
> Your feedback on this has been that you will **update the figure to show the ground truth for the right hand side particles**. However, in Fig. 19 you state that **no ground truth is available because the authors of GNS did not provide the physical parameters** for the simulation. Now, if you could have simply added ground truth initially to defend the claim of generalisability, why didn't you do so?
>
> I remain skeptical on this point.
>
> ---
>
> **2. On Fig. 6**
>
> The authors clarify that the processor is invariant to the number of particles, but not the whole pipeline. However, in line 469 the subject of the sentence is NeuralMPM, which I believe refers to the pipeline. In Fig. 6, authors point out that the **FPS remain stable longer** (i.e., it is more robust as the number of particles increases). This, however, is fundamentally different to invariance, and I believe the claim is overstated.
>
> ---
>
> **3. On Fig. 4**
>
> I appreciate the use of a different dataset. If DamBreak 2D or Variable Gravity show such significant improvement, it is important to show that.
>
> ---
>
> **On the comparison with GINO**
>
> I asked for a comparison with Li et al. 2024, but maybe I should have clarified that I was referring to the GINO paper. The authors say:
> *Li et al.'s method is a hybrid classic-ML approach. As such it benefits from the strong inductive biases of the physics of the method, but its range of applicability is smaller.*
>
> This doesn't apply to GINO, hence I believe there has been some confusion on this? What paper were you referring to? The authors continue by stating:
>
> *On the other hand, NeuralMPM is a fully data-driven emulator, and, as such, it can be applied to a wider range of problems at the price of less powerful inductive biases.*
>
> This claim actually applies to virtually any ML based emulator? I don't see this as a novelty of NeuralMPM.
>
> ---
>
> **On the intuition on the superiority over FNOs**
>
> The author state that FNOs perform worse due to the sharp discontinuities in the input density and output velocity fields. Is this your intuition or can you provide references on this? Also, NeuralMPM discretises particles into voxels, which I assume, from Fig. 1, are fully filled? (i.e. no voxels without particles?).  Regardless of a grid fully filled/sparse with voxels, surely your U-Net suffers from the same issue as a FNO?
>
> I remain skeptical on this.
>
> ---
>
> ### Summary:
>
> I feel the authors have only partially addressed my concerns, leaving me with some doubts explained in the points raised above. Specifically, the lack of clarity on the generalisability claim and GT of the rectangular domains, the lack of proof of the advantages over the proposed U-Net pipeline over FNOs, the lack of comprehensive benchmarking (the authors have brought the compared models from 2 to 3) and the lack of improvement over the reported GNS pipeline (Sanchez 2020) leave me skeptical.
>
> Since I still have doubts on the novelty, impact and on some overstated claims after the rebuttal, I feel that 5 is an appropriate score for the manuscript. I prefer to wait for the final version of the manuscript to finalise the score, to see if the edits in the text make a more compelling argument, especially for Fig 4, 7 and 19.

---

> ### Author Response · Authors · 2024-11-27
>
> Thank you for answering, engaging, and clarifying your opinion.
>
> -------
>
> 1. In the supplementary material of the paper we have included videos of the rollouts in Fig. 7, showing 10 WaterDrop-XL (larger number of particles) rollouts, and the rectangular rollouts presented in Fig. 7 and Fig. 19. As we said, we cannot add the ground truth used for this figure as we do not have access to the same simulator, physical parameters, or even the generalization datasets from the GNS paper (we contacted the authors and asked for them, but they said that open sourcing the generalizaion datasets was outside of the scope of the project). What we proposed was to generate a new dataset with a simulator to which we have access, and do the experiments on that. We will not be able to do that within the rebuttal period, as our access to compute is limited. However, we once again stress that the videos of Figs. 7 and 19 are already in the supplementary material, showing the full generalization rollouts.
>
> -------
>
> 2. We will revise the claim of invariance to clarify it is for the processor. However, we note that our method is orders of magnitude more memory efficient than the baselines, and ask that this be taken into account. Further, both p2g and g2p scale linearly with the number of particles.
>
> -------
>
> 3. We have updated Fig 4. and added similar figures for other datasets (including the old Fig. 4) in the appendix. Further, we have included videos of rollouts on all datasets of all baselines, NeuralMPM, and the ground truth that visually shows the difference. Further, in Figs. 2 and 3 of the rebuttal
> PDF (9 and 10 in the appendix of the paper) we show the rollout error (MSE and EMD) as a function of time for all datasets (including DamBreak2D and VariableGravity) and baselines. The advantage of NeuralMPM is clear in these plots.
>
> -------
>
> GINO: Thank you for the clarification and sorry for the confusion. GINO introduces a NO on irregular grids thanks to a GNN-based learned transformation from irregular to regular grids. Their work within the context of Eulerian simulations is impressive. The main difference between our two methods is the context, we work on a dynamic point cloud, requiring graph reconstruction at every time step. This requirement is one of the limitations of graph-based simulators and one of the reasons for GNS slowness and memory inefficiency compared to NeuralMPM. On the other hand, NeuralMPM has linear particle aggregation complexity. GINO does this via a linear O(N) method, similar to NeuralMPM. However, our irregular to regular mapping (\textbf{p2g}) does not have any learnable parameters and requires no training. Moreover, we have applied NeuralMPM to Lagrangian point clouds, and have shown its advantages to other emulators that target the same domain. The DMCF baseline uses Open3D for the graph construction (same as GINO), and we have shown to be faster and more memory efficient than it.
>
> The claim about NeuralMPM and inductive biases, as you say, is true for almost all ML-based emulators. This is what we meant by "NeuralMPM is a fully data-driven emulator". We do not claim to have introduced this advantage, rather it drives from being data-driven. Hybrid simulators, which we discuss in L132-137, do not fully enjoy this advantage.
>
> -------
>
> FNO: We want to highlight that the particular choice of processor is not our main contribution, rather the whole pipeline is. In our experiments, we found the UNet to be better than the FNO but that does not mean NeuralMPM must always be used with a UNet processor, nor do we claim the UNet to be a better architecture than the FNO in general.
>
> That said, we will give our intuition for why the UNet performed better in our experiments. The voxels that have no particles are empty, and therefore have 0 density and 0 velocity. The FNO operates only on lower frequencies, removing the higher ones (Fourier Neural Operator for Parametric Partial Differential Equations, Li et al. Section 4 and Fig. 2). This can make modeling sharp discontinuities hard. That is simply our intuition, and we do not have a reference for it. We once more highlight that the comparison of a UNet and an FNO processor is not the main contribution of our work, nor is it generally applicable to all problems.

---

> > ### Author Response · Authors · 2024-11-27
> >
> > ----------
> >
> > We would like to address specifically your claim of the lack of comprehensive benchmarks. We have reported the results of 17 models (6 NeuralMPM, 6 GNS, and 5 DMCF) after spending more than 60 GPU-days on fine-tuning GNS and DMCF (more than 120 GPU-days total), which is much longer than we spent fine-tuning NeuralMPM. We have presented generalization results for NeuralMPM on 2 datasets, for only one of which we do not have the ground truth. We have videos for both generalizations in the supplementary material and have added videos for all other experiments and baselines, together with Figs. 9 and 10 in the appendix.
> >
> > We will also address specifically the claim of lack of improvement over the GNS pipeline. NeuralMPM is much faster, taking ~6% of the training time of GNS (15h vs 10 days, Fig. 5) for a superior performance on most datasets, and close performance on the other two (Table 1). Further, it uses 10x-100x less memory (Fig. 6, they use a batch size of 2 while we can go to 64 or 128), and has 4x faster inference time (Fig. 6). We use the implementation of the original authors both for GNS and DMCF (line 363), together with the hyperparameters they provided where possible. In new datasets, we performed extensive hyperparameter searches on both (60 GPU-days for each, more than we did for NeuralMPM).
> >
> > Following your previous advice we have also reformulated the abstract and conclusion to be more quantitative. We hope this clarifies our position and addresses your concerns. Thank you again for the constructive feedback.

---

> ### Comment · Reviewer_DPie · 2024-11-27
>
> Following the author's response, and having noticed the changes provided to the manuscript, I decide to retain my score of 5. The motivations behind this choice below:
>
> **Fig. 4.** I appreciate you changing it to a more descriptive dataset and adding further examples in the appendix. However, I feel the original improvement over other datasets is slightly overstated: while VariableGravity shows a clearer improvement with respect to other methods, it seems not that different with respect to the original WaterRamps. Results in Table 1 seem misleading, considering that, while the MSE is much lower, visually there's little difference. I have looked at some of the video comparisons, too, for the WaterDam 2D dataset, and improvements seem marginal.
>
> **Fig. 3.** Have you retrained the GNS? You say that *the dotted orange line (2.4 ×10−3) indicates the MSE we obtained for GNS after 240 hours (20M training steps).*. If I am not mistaken, in the previous version of the manuscript you had reported the results *they* obtained in orange, which were consistently lower than your ablations (and that you have removed in the revised manuscript), and in a different colour you reported your re-implementation. The original version of the manuscript could have hinted at the fact that GNS, if reimplemented, might not perform as well as originally reported, but by further tuning you could reproduce those results, since the orange line now is much closer to what originally reported.
>
> **Generalizability**: I feel the authors haven't tackled this issue. I have looked at some of the WaterDrop-XL dataset videos, and I feel they confirm my initial skepticism. There's some artifacts that don't seem realistic physically, such as water blobs deforming as they come down or that they "slide" along the side of the domain. Now, this dataset is not that different from those presented, as the number of particles is larger but the domain is similar. What is happening in Fig. 7-21-22 is likely a similar phenomenon - the behaviour is quite unrealistic, especially considering that the domain is much more different than those of training data. I think these observations are quite obvious to an external person, since such figures simply don't show a believable evolution of the fluid with time. Considering the lack of ground truth, the claim of generalisability is, at best, overstated.
>
> I appreciate that conducting further experiments might be hard. I was not expecting that, but rather an honest explanation of that behaviour.
>
> **On benchmarking**: I feel you're being imprecise. You have reported results of 17 *configurations* of 3 *models*, one of which is your own, and one of which has been published 4 years ago and, based on the original Fig.3, seems to still outperform NeuralMPM. Now, implementing 17 different models is a different thing than 3 models with different parameters, and it does not require the same amount of effort. I would have been happy simply with a deeper search of the literature.
>
> I understand that NeuralMPM has other advantages over GNS, and I appreciate the efforts put in the rebuttal, including the authors' clarification on GINO and FNO. However, several claims made, including the generalisability over more particles/rectangular domain, the large improvement over past methods and the invariance with the number of particles, seem quite opaque, even after the revision.
>
> Having considered these facts, the lack of extensive benchmarking and of solid novelty, I would consider this paper below the acceptance threshold. I believe this is fair and conservative, also considering other's reviewers harsher opinions on this work.

---

> > ### Author Response · Authors · 2024-11-28
> >
> > Thank you for your prompt reply!
> >
> > -------
> >
> > **Regarding the implementation**
> >
> > As stated in L366 in the paper, **we have not reimplemented GNS**, instead **we have used the implementation and hyperparameters provided by the original authors, Sanchez-Gonzales et al.**. In the previous manuscript, we had the results they reported, which we were not able to reproduce after numerous tries **with *their* implementation and parameters**. Instead, the orange line shows the **best** model we obtained with their implementation. Please also note that we have **improved** the best GNS on MultiMaterial, having an MSE of $14.79$ instead of $16.9$, which is what the original authors reported.
> >
> > In addition, in the appendix (Tables 2 and 3) we compare against another, more recent (LagrangeBench, Toshev et al. 2024) implementation of GNS and SEGNN, with the same results. NeuralMPM outperforms both.
> >
> > -------
> >
> > **Regarding the argument of little improvement**
> >
> > We have presented the quantitative results as they are, showing the improvements of NeuralMPM on most of the datasets.
> >
> > We further remark that NeuralMPM has better EMD across all datasets and the EMD might be a better metric to measure the quality
> > of a fluid simulation. Additionally, GNS performs quite poorly qualitatively on MultiMaterial compared to NeuralMPM, as shown in the videos in the supplementary and the drive. On MultiMaterial, our GNS is **better** than the one reported by Sanchez-Gonzalez et al., and NeuralMPM is still better. More importantly, NeuralMPM trained in 6% of the time of GNS, is significantly faster and consumes orders of magnitude less memory. Even if we did not achieve comparable or superior accuracy for MSE and EMD, which we do, the speed and memory efficiency of NeuralMPM are a significant advantage.
> >
> > -------
> >
> > **Regarding generalizability**
> >
> > Thank you for acknowledging the challenge involved in conducting new experiments. We have revised the paper again and included Figure 11, showing the MSE on WaterDrop-XL for NeuralMPM and GNS. Both models were trained on WaterRamps, and on WaterRamps the GNS model had a slightly better MSE.
> > Despite that, NeuralMPM is better at this generalization task, as shown in the figure. We will include in the videos on drive the qualitative comparison of the two models on WaterDrop-XL, and inform you when it is ready (as it takes some time to render). Further, and for the same reason, we will include
> > in the final version of the paper Figure 12, a figure identical to Fig. 11 but showing the EMD instead of the MSE. We have not done so yet because the EMD is a time-consuming metric to compute, but we will do so for the final version. From the simulations that have finished computing, we see NeuralMPM having
> > lower EMD than GNS across all of them, in line with the MSE from Fig. 11. Again, **this GNS model was trained with the implementation and hyperparameters provided by the original authors**
> >
> > As for the explanation of NeuralMPM behavior on WaterDrop-XL, this is our intuition:
> > The processor in NeuralMPM is not aware of the particles, it sees only grids, so, by definition, it is invariant with respect to their numbers. On the other hand, it "sees" densities and velocities, which are physical quantities characterizing certain dynamics. While WaterDrop-XL looks similar, its density is far larger than that of WaterRamps, and so is the velocity of the particles. We are therefore in a different dynamical regime, which poses a generalization challenge, and is the reason behind NeuralMPM having some artifacts. However, in the videos you will see that GNS has much worse artifacts.
> >
> > -------
> >
> > **On benchmarking**
> >
> > We have trained 17 different experiments, 6 for NeuralMPM, 6 for GNS, and 5 for DMCF. The original Fig. 3 showed that NeuralMPM outperformed GNS for for a large number of hyperaparameter configurations, but not for all of them (an unreasonable expectation). In Table 1 and Figs. 9 and 10 it is shown quantitatively that NeuralMPM outperforms GNS. In Figure 11 it is also shown that NeuralMPM does better at the generalization problem, as will also be shown in the future videos and Figure 12.
> >
> > In the experiments, the datasets have different physical properties, including multi-material interactions, variable gravity, and complex boundary conditions. It is an exhaustive benchmark, superior to, for instance, the one carried out by DMCF.
> >
> > -------
> >
> > To summarize, **we have used the GNS implementation and hyperparameters of the original authors** and NeuralMPM outperformed GNS in terms of EMD across all experiments, and in terms of MSE across all of them but two, on which it was close. In addition to training in 6% of the time, being 10x faster, and having a memory footprint 100x smaller, NeuralMPM showed better quantitative and qualitative results on the generalization task of WaterDrop-XL, despite using a supposedly better GNS (the one that has lower MSE). A task that was also used to test generalization in the original GNS paper.

---

> ### Comment · Reviewer_DPie · 2024-11-28
>
> I apologise if I am repetitive, but I'll mention this a fourth time:
>
> Current Fig. 23, rows 2, 3, 4, 5. Does that seem a believable evolution of Snapshot 1 given the Initial Condition on the left? Given your answer to this question, and considering that no ground truth is available, can you claim generalisability?

---

> ### Author Response · Authors · 2024-11-28
>
> We have provided videos of the full rollouts of Figure 23 in the supplementary material ("Rectangular" folder, both in the zip file and in the drive mentioned in the general comment, link here  https://drive.google.com/drive/folders/1lj8a-iHoAXOyT3HFl5mXa0Bc5_nHbdZb). They look realistic and believable, which is why we claim generalizability.
>
> In the same zip and drive, under WaterDrop-XL we provide rollouts for another generalization task, one which was also used in the original GNS paper. We train the model on WaterRamps and do the test rollouts on WaterDrop-XL, a set with more particles, more energy (i.e. higher velocities), higher density, and longer simulation. We beat a GNS model trained and used in the same way (Figure 11) in terms of the MSE. We will also provide full rollout videos for NeuralMPM and GNS on this dataset, where the advantage of NeuralMPM is clearly seen.
>
> In the supplementary material (zip and drive) we have provided full rollouts for all sims on all datasets with the ground truth, NeuralMPM, DMCF and, GNS shown together. The figures in the paper are snapshots from these videos, but the videos provide a much more detailed context and they need to be taken into account.

---

> > ### Comment · Reviewer_DPie · 2024-11-28
> >
> > I have seen the videos and I have to acknowledge my mistake - I assumed blobs of water would just fall down under gravity, while the initial condition actually implies some initial velocity and direction of the movement. This makes Fig. 23 reasonable and a thr claim of generalizability likely.
> >
> > However, based on the reasons above, I still feel that 5 is a reasonable score.

---

> > > ### Author Response · Authors · 2024-11-28
> > >
> > > Thank you for acknowledging the mistake. It would be constructive and helpful if you could tell us, in light of the arguments and evidence from our answers, what exactly are those reasons. Which of your concerns remain unanswered?

---

> > > > ### Comment · Reviewer_DPie · 2024-11-28
> > > >
> > > > No concerns remain unanswered.
> > > >
> > > > The lack of extensive benchmarking (comparison with just 2 models, DNS and DMCF), marginal improvement over DNS (Fig 3) and of solid novelty (extension of MPM), as well as some claims that seem overstated (generalisability, since WaterDrop-XL doesn't seem realistic to me and I don't see why a large number of particles makes such a big difference in the shape of the blob coming down, invariance to number of particles, improvement over DNS) make me doubtful of the contributions of this work.
> > > >
> > > > However, I do acknowledge that I have mentioned that I would be increasing my score if Fig. 23 and Fig. 4 were better addressed, and you did. I am not particularly impressed with what was shown, but I will keep my word and raise my score to 6.
> > > >
> > > > I appreciate your patience during the rebuttal.

---

> > > > > ### Author Response · Authors · 2024-11-29
> > > > >
> > > > > Thank you for keeping an open mind and reconsidering your score, we appreciate it.
> > > > >
> > > > > To summarize, we believe that our contribution is solid, since we have a significant improvement of the training time, inference time, and memory consumption over GNS and DMCF. For WaterDrop-XL, the main difference is the different density and total energy in the simulation change dynamics to be out of distribution. Further, GNS has worse results on WD-XL, despite the model being supposedly better.
> > > > >
> > > > > Again, thank you for your engagement.

---

### Official Review · Reviewer_MkVt · 2024-10-30

**Soundness:** 2
**Presentation:** 3
**Contribution:** 2
**Rating:** 5
**Confidence:** 4

**Summary:**

This paper proposes combining conventional MPM simulation with neural networks to enhance its training speed and generalization ability. This method interpolates the Lagrange particles into the background grid and uses an image-to-image network to process and solve motion equations to obtain the final outcomes.

**Strengths:**

Strength
1. Replacing parts of the traditional numerical simulation framework with neural networks instead of learning the whole dynamic process can improve the generalization ability, reduce the training difficulties.

2. The hybrid physical solver can be considered the next generation of physical simulation because the classic part and neural part can mutually benefit from each other. The classic part provides basic priors and genralization ability and the neural part improve the accuracy and efficiency.

3. Conventional GNN-based particle simulators requires large amount of data for training, while the proposed method only need fewer training data thanks to its MPM solver priors.

**Weaknesses:**

Weakness:

1. This work relates to hybrid (classic and learning-based) simulation solvers, but it seems to provide insufficient background on related work in the area, especially lacks more recent works. Most of the works mentioned in the paper are from 2023 or earlier. This will make the paper appear less to-date, even though the method is innovative. Hence I list some relevant recent works below and suggest the author can discuss and compare with them to make the background sections to-date.


"DEL: Discrete Element Learner for Learning 3D Particle Dynamics with Neural Rendering." 2024. This paper also builds a hybrid numerical solver by integrating conventional particle mechanics into neural network design, which should be included in the same subfield with NeuralMPM, i.e. the neural-based physics solver.

“A Neural Material Point Method for Particle-based Simulations” 2024  By the way, this paper also proposes a hybrid model based on the MPM method, could you please clarify what is the main difference between NeuralMPM and this one?

"ElastoGen: 4D Generative Elastodynamics". 2024. This paper introduces elastic dynamics into the learning generative pipeline, it directly generates the network parameters that are designed according to projection dynamics, which seems to be an application of hybrid solvers.


2. NeuralMPM still requires full 2D particle trajectories to train the model, which could be hard to access in most cases.  Moreover, This method is currently limited to simulating 2D scenarios. Are there any obstacles to extending it to 3D? I believe this could be achieved by simply replacing the UNet with its 3D convolutional version.

3. I hope the author can provide a quantitative analysis of this method's execution efficiency compared to traditional methods. Since the process is carried out by a neural network, efficiency is expected to improve, which could be another potential advantage of this approach.

4. Explicit integration in traditional methods is very sensitive to the time step size. After training, I’m curious whether this method can perform computations with a larger time step. Compared to explicit integration in traditional methods, can it operate with a larger time step?

I will adjust my score according to the author's feedback.

**Questions:**

See Weakness

---

> ### Author Response · Authors · 2024-11-27
>
> Thank you for your review and comments, we will address them below.
>
> 1. The works you have suggested are contemporaneous with ours, so a comparison is naturally impossible. They are hybrid methods, on which we have commented in L132-137. In summary, these approaches achieve impressive results by leveraging extensive physics knowledge, but this reliance also limits their applicability. Fully data-driven methods, like NeuralMPM, are more general and can be applied to a wider range of problems.
>
> 2. Yes, you are correct. Replacing the processor (in our case a UNet) with a 3D counterpart will extend the method to 3D. We have some early results for that that we will publish in a follow-up paper.
>
> 3. We have provided such analysis in Fig. 6, showing that NeuralMPM is faster and more memory efficient than the baselines and the ground truth MPM simulator.
>
> 4. Time integration is controlled by the time bundling $m$ parameter. In Fig. 3 we study the effect of $m$ on the accuracy of the model. It can be seen there that it can operate at a time step 8x-16x that of the ground truth data. In turn, the ground truth simulator operates at a much lower time step, making this advantage even more pronounced.
>
> We hope to have addressed your concerns and that you therefore will consider raising your score. Thank you for your time and effort in reviewing our work.

---

### Official Review · Reviewer_GJwq · 2024-11-04

**Soundness:** 1
**Presentation:** 2
**Contribution:** 1
**Rating:** 3
**Confidence:** 4

**Summary:**

This paper proposes to adopt material point method for particle-based simulation. The method first interpolates particles onto the grids, and simulate the interactions among grids to generate velocities, which are further distributed to particles again. The proposed method delivers faithful rollouts as shown in experiments.

**Strengths:**

* This method tries to integrate with MPM for simulation.
* The approach seems to be more robust in specific domains.

**Weaknesses:**

1. The method is technically limited.
    1. The adaptation of MPM is naive. The main difference between the proposed method and the original MPM is that the proposed method does not need the template states in material space, or equivalently, the “material space” in this method is dynamically determined. For example, when predicting frames from $t$ to $t+m$, the “material space” is defined by states at time $t$. In this way, this method is very close to existing MPM pipeline, leading to limited novelty.
    2. The continuous predictions of m steps seems questionable. For example, when the particles move very fast from $t$ to $t+m$, the grid constructed at time $t$ may not be suitable for particles at time $t+2$, which is in the middle of the duration, because the distribution of the particles changes dramatically. In other words, such formulation seems to be more reasonable for particles moving very slow, where the distribution of particles changes slowly as well or remains similar, leading to limited fields of application.
2. There are some questions about the data.
    1. The data is a bit tricky. Since the data is also generated by MPM methods, the data will have similar bias as the proposed model. Both the data and the proposed method naturally behave similar under the framework of MPM. In other words, if the data is generated by other physics simulators without MPM, such as PBS, the proposed method may fail to learn the dynamics.
    2. How many particles are included in each scenario? And how long is each sequence?
3. The evaluations are insufficient.
    1. The author only provides results on 2D domain. Can the method be applied to 3D domain as shown in the GNS method? The settings of only using 2D domains seems a bit simple.
    2. As shown in the generalisation results in Appendix, the author claims that the missing of ground truth results from the unknown simulation parameters. However, since the method can be applied to simulate different materials as mentioned at L230-231, the author can adopt a set of reasonable but simple parameters and generated the corresponding scenarios, enabling quantitative comparisons on those domain. Existing evaluations are insufficient to proof the generalisation abilities of the proposed method.
    3. In the supplementary videos, the visualisation results obtained by baselines are missing, which prevents from comparing with baselines qualitatively. Since the work focuses on dynamic predictions, the video results of animations are also important to evaluate the methods.
    4. At L025, the author claims that the proposed method can better handle long-term predictions. However, from existing evaluations, one is hard to find the evidence to prove this claim. Can the author explain clearly or even provide more results for long-term predictions?
4. The author may considers TIE [1] as related work or even compare with it, which is also a particle-based simulator.

[1]. Yidi Shao, et al. Transformer with Implicit Edges for Particle-based Physics Simulation, ECCV2022.

**Questions:**

Besides questions mentioned in weaknesses, here is one more question:
1. Why does GNS take so long to train as mentioned at L252? Is it because different training schedules are applied? For example, the training of GNS adopts much smaller learning rate or batch size?

---

> ### Author Response · Authors · 2024-11-15
>
> Thank you for your review. We have addressed your questions and concerns to the best of our ability below. While we value your feedback and appreciate constructive critique, we find it difficult to identify clear actionable steps based on the points raised. Could you clarify the reasoning behind the rejection and whether our explanations below have influenced your perspective?
>
> ----
> For the weaknesses
>
> 1) - Our purpose is not to develop a MPM competitor but rather to build a simple learnable model, inspired by the structure of MPM in order to inherit some of its advantages. The model does benefit from the MPM pipeline (particles -> grid -> particles) yet remains simple. We have adapted some of the solutions used in MPM to problems in the neural emulation field (e.g. the problem of extracting information from the point cloud L57-L62, L107-L110), being novel to the best of our knowledge.
>
>    - The time bundling parameter $m$ is a hyper-parameter which can be tuned, depending on the training data. $m$ can be set to 1, leading to a 1-step-forward autoregressive model. Ablations of this hyper-parameter are shown in Fig. 3 and Section 4.1
>
>
> 2) - We note that we did not generate most of the data, and instead used the datasets provided by Sanchez-Gonzalez et al., as they represent a broad range of physical interactions. Further, we also have an SPH dataset (Dam Break 2D), where NeuralMPM outperforms the baselines. To further show we are not limited to the MPM domain we will add another non-MPM dataset.
>
>    - The number of particles for each dataset is reported in Table 1 (second column, "N"). Note WaterDrop-XL goes up to 10k. The third column is the temporal duration in the number of frames. The duration in seconds (i.e., depending on dt), can be found in the original dataset references. The Dam Break 2D dataset uses dt=0.03 (i.e., 12s-long), while the all other datasets use dt=0.0025 (i.e., from 1 to 2.5s-long), which is often long enough to reach a stable state, considering the velocities.
>
>
> 3) - We strongly disagree that the evaluations are insufficient. We train the model on 6 different dataset (WaterRamps, SandRamps, Goop, MultiMaterial, VariableGravity, Dam Break 2D), comprised of different materials and dynamics, and compare its performance on those dataset against well-established data-driven baselines. Further, we evaluate generalization quantitatively and quantitatively (WaterDrop-XL) on one dataset and qualitatively on another (Large rectangular domains). We also showcase that the model’s differentiability can be used to solve inverse problems. Moreover, the method can be trivially extended to 3D domains, by replacing the processor with a 3D counterpart. However, 3D training requires more memory and training time (compared to the 2D case), making extensive experiments very time consuming, for both NeuralMPM and the baselines. For example, we could not even run the GNS code with 3D data on a high end GPU (Telsa V100, 32GB).  We have preliminary 3D results, and plan to present them in a future work, but believe they add little to the discussion here.
>
> 4) We acknowledge that evaluating against ground truth is feasible but was not done mainly to reuse the same model (i.e., trained on WaterRamps) as used for WaterDrop-XL. We agree this is not ideal, and will create new simulations that will be extended the same way, with corresponding ground truth.
>
> 5) We thank the reviewer for this good remark, we will add videos for comparing against the baselines visually.
>
> 6) In L25 we said we achieve “comparable or superior long term accuracy” compared to other methods, namely GNS and DMCF.. This is shown in Table 1, where NeuralMPM has lower MSE on almost all the taks, and lower EMD on all of them.
>
> 7) Thank you for bringing this work to our attention. We will add it in the relevant work section.
>
> ------
> As for the question
>
> 1) GNS indeed adopts a much smaller batch size, as only a size of 2 fits on a V100 GPU with 32GB of RAM. In addition to that, the method is slower due to the graph construction, which becomes quite costly for increasing numbers of particles. In contrast, NeuralMPM, being more memory efficient, can fit much larger batches in memory, and bypasses the expensive graph construction.
>
> -------
>
> We thank you again for your review, and have proposed the following changes to address your concerns:
> - Add another non MPM dataset.
> - New results showing the generalization to different domains, with ground truth.
> - Add comparison videos for the baselines
>
> In light of our answers and the proposed changes, would you raise the score, or provide additional actionable input and elaborate on how you have arrived at the decision to reject?

---

> > ### Comment · Reviewer_GJwq · 2024-11-26
> >
> > Thank you for your reply and I appreciate the author's efforts to clarify the questions. However, I would keep my score mainly for the following reasons:
> >
> > 1. The technical novelty is not well explained. In other words, the proposed method mainly replaces the grid update by neural network comparing with the traditional MPM pipeline, leading to limited novelty.
> > 2. While existing methods (e.g. GNS) are able to deal with 3D cases, the experiments in this paper focus on 2D scenes, leading to the concerns of the method’s abilities and effectiveness.
> > 3. From my experience, the training time for GNS is not as slow as mentioned in the paper, even with around 10K particles. To achieve fair comparisons, the author should implement the GNS with similar coding technique, such as the deep learning framework, and similar hyper-parameters, such as the batch size.

---

> > > ### Author Response · Authors · 2024-11-27
> > >
> > > Thank you for your comments and for engaging with us in the discussion!
> > >
> > > 1. The technical novelty should be compared against the baselines, not MPM. We have not claimed to introduce a new MPM version, but rather to have strongly outperformed the baselines. We do not believe a 6% training budget, 5x-10x faster inference time, and 10x-100x less memory usage can be disregarded.
> > >
> > > 2. 3D UNets and Neural Operators are used extensively in the literature. Simply using the appropriate processor architecture would lead to a 3D method. There is no reason why NeuralMPM could not be used in 3D.
> > >
> > > 3. We have used the GNS implementation from the original authors, Sanchez-Gonzalez et al. (2020). It is beyond the scope of our work to improve and optimize their implementation. Further, in the appendix we added a comparison against a different implementation of GNS and SEGNN (Brandstetter et al. 2021) provided by LagrangeBench (Toshev et al. 2024). The results remain the same.

---

### Official Review · Reviewer_w7nQ · 2024-11-04

**Soundness:** 3
**Presentation:** 3
**Contribution:** 2
**Rating:** 6
**Confidence:** 3

**Summary:**

Inspired by material point method (MPM), the authors proposed a neural emulator framework for particle-based simulations. The downstream tasks include weather forecasting, computational design, inverse problem, and animation industry. Their method is compared to the traditional MPM, and two other state-of-arts data driven methods, namely DMCF and GNS.

**Strengths:**

- The article is easy to follow, and sufficient background knowledge is provided to readers. I appreciate that.
- The proposed neural emulator is demonstrated more efficient than traditional MPM and other data-driven methods in computational time, more accurate, and it requires much shorter training time, and smaller memory usage.
- Since the model operates on regular grid only, it allows use of a simple network architecture such as convolutional UNet.
- Being able to solve inverse problems to predict boundary condition is an interesting idea.

**Weaknesses:**

- The common drawback of autoregressive models is its instability for long-term predictions. The examples shown in the paper all seem to have a short period of time simulated. Although rollout loss can mitigate error accumulation to some degree, it cannot help long-term prediction. If a simulation (for example consider water particles continuously added through an inlet)  lasts much longer, the model might not be able to predict accurately after certain time.
- From my understanding, for a different grid resolution, whether coarser or finer, it would require training a new model.
- Based on the current memory usage and training time, it might be difficult to scale to 3D problems with the current setups.

**Questions:**

- How is `taichi-MPM` implemented? Also consider comparing to state-of-arts MPM implementations, such as [this one](https://dl.acm.org/doi/10.1145/3386569.3392442).
- The paper claims to be able to emulate up to 30 millions particles, but I do not see any examples of videos or snapshots.
- About *able to generalize to different domain shapes* on line 188 page 4,  what exactly are "different domain shapes"?  Does it work for non-square shapes like circles?
- Do you have examples on smoke simulation such as smoke passing an object? Have you tried on examples with nonzero initial velocities?
- Does it require separate models for each example, such as `WaterRamp`, `SandRamps`, `Goop`? Can you train a single model that generalizes to problems with any different setups, ie, fluid property, boundary condition, initial velocity?
- Can you explain why numerical simulators are non-differentiable?
- It would be nice to have a diagram on network architecture.

---

> ### Author Response · Authors · 2024-11-15
>
> Thank you for taking the time to review our paper.
>
> ----
> Concerning the weaknesses section:
>
> 1) s you say, autoregressive model stability is a challenging problem. Addressing it is an active field of research on its own right [1] We chose to decouple the framework used for the neural emulator from the rollout stabilisation mechanism by using a common technique, autoregressive training [1, 2]. In Section 4.3 (L470-L478) we produce longer rollouts of 1000 timesteps. To strengthen this argument we will provide a video of longer simulations where we add water particles.
>
> 2) That depends on the architecture of the processor used. Both the U-Net and FNO backbones can process different grid sizes without retraining. As shown in Fig. 7, the U-net backbone can extend to larger domains (and thus different grid sizes), but might fail at a finer grid in the same spatial extent, due to fixed kernel sizes. However, using an FNO backbone would support updating grid velocities over different grid resolutions without retraining, thanks to its operator nature [3]
>
> 3) While it is true that 3D requires more memory and training time, we already have preliminary results that are less expensive than other data-driven methods, especially GNS which we could not retrain on 3D data on 32GB-VRAM GPUs. We plan to present those results in a future work, as we believe they do not add to the discussion here, and require time consuming experiments.
>
> ----
> Answers to questions:
>
> 1) We use the reference implementation provided by Taichi (see https://github.com/taichi-dev/taichi/blob/master/python/taichi/examples/simulation/mpm128.py), based on MLS-MPM [1]. We want to stress the fact that our method is not a direct comparison to MPM, but rather a data-driven approach, where MPM (and SPH), was used as ground truth. Our goal is not thus to compare to MPM implementations.
>
> 2) In Fig 6. we limit-test NeuralMPM, the baselines, and the ground truth simulator Taichi-MPM in terms of speed and memory. We did not test the accuracy of the simulations for any of the methods.
>
> 3) This line refers to rectangular domains, such as a larger non-square domain. A circular domain can be achieved by inscribing the circle in a square and placing the appropriate boundary conditions.
>
> 4) Most of the simulations have non-zero initial velocities, this can be seen in the videos of the supplementary material. We do not have smoke simulations.
>
> 5) - We trained a different model for each dataset. However, the model trained on MultiMaterial can handle simulations from WaterRamps, SandRamps, or Goop. The MultiMaterial dataset itself contains single-material simulations.
>    - Different properties can be embedded into the model, such as different fluid types (e.g., Multimaterial), but the model will be limited by its training data, except if the physical parameter is embedded into the model directly, like we did in VariableGravity.
>    - Boundary conditions are naturally handled through the use of static "wall" particles for borders and obstacles. This allows for very general boundary conditions.
>    - Initial velocity is also naturally handled as NeuralMPM is conditioned on a single previous state. Most of the simulations have non zero initial velocities.
>
>
> 6) To be differentiable, the whole implementation (i.e. each operation) must be differentiable, and simulators are complex and expensive to craft, sometimes requiring reformulations of the dynamical models themselves. Further, the computation of the derivative has to be efficient, which is a non-trivial challenge for complex or long simulations.
>
>
> 7) Figure 1 depicts the end-to-end architecture of the network, from the input point cloud to the m output point clouds. The processor part (i.e., the neural network) is not depicted in detail as any image-to-image backbone can fit, and we do not consider this to add meaningful information to the description of the pipeline.
>
> ----
>
> We thank you again for your review, and have proposed the following changes to address your concerns:
> - Longer simulations where particles are added.
> - New results showing the generalization to different domains, with ground truth.
> - Add comparison videos for the baselines
>
> Having taken the perspective provided by our answers, and the proposed changes into account, would you kindly consider raising your score, or provide us with further actionable feedback?
>
>
>
> -----
> References
> [1] List, B., Chen, L., Bali, K., & Thuerey, N. (2024). Differentiability in Unrolled Training of Neural Physics Simulators on Transient Dynamics.
>
> [2] Prantl, L., Ummenhofer, B., Koltun, V., & Thuerey, N. (2022). Guaranteed Conservation of Momentum for Learning Particle-based Fluid Dynamics. ArXiv, abs/2210.06036.
>
> [3] Li, Z., Kovachki, N.B., Azizzadenesheli, K., Liu, B., Bhattacharya, K., Stuart, A.M., & Anandkumar, A. (2020). Fourier Neural Operator for Parametric Partial Differential Equations. ArXiv, abs/2010.08895.

---

> > ### Comment · Reviewer_w7nQ · 2024-11-18
> >
> > Thank you for your response.
> >
> > I see that neural emulators could have more potential uses in simulation, such as solving inverse problems. The idea is interesting, but at this stage, it is lacking practical use, since it is not ready to tackle 3D problems yet.
> >
> > Thus, I decided to maintain my score.

---

> > > ### Author Response · Authors · 2024-11-27
> > >
> > > Thank you for your comments.
> > >
> > > 3D UNets and Neural Operators are used extensively in the literature. Simply using the appropriate processor architecture would lead to a 3D method. There is no reason why NeuralMPM could not be used in 3D.

---

### Official Review · Reviewer_sSsx · 2024-11-05

**Soundness:** 2
**Presentation:** 3
**Contribution:** 2
**Rating:** 3
**Confidence:** 4

**Summary:**

In this paper, the authors present a Neural Material Point Method (NeuralMPM) for improving the efficiency of the numerical simulation. Comparing to traditional MPM method, NeuralMPM  perform grid operations using image-to-image neural networks. The proposed method is validated on several fluid and fluid-solid interactions datasets. It demonstrates that the proposed method can effectively reduce the training time while achieve comparable or better long-term accuracy, comparing to the benchmarks.

**Strengths:**

The paper is well-organized and easy to follow.

**Weaknesses:**

The paper describes computational fluid dynamics (CFD) as the background or motivation for NeuralMPM. MPM is powerful for simulating objects with large deformation (elasticity, plasticity, fractures etc) and handling collision contact naturally. However it may not the best common choice for simulating incompressible fluid (such as water) in the general CFD context, as it is not handling the divergence-free constraint in incompressible fluid. In these scenarios, another hybrid Eulerian-Lagrangian method FLIP, or pure Lagrangian based method (such as SPH) might be better choices.

The original MPM pipelines can be generally divided into three stages as particle-to-grid(p2g), grid-update, grid-to-particle(g2p). The proposed method essentially replace the original grid-update with a U-net and utilize p2g and g2p as is. Therefore, the main claim (From Line 519 tp Line 521 "The use of voxelization allows NeuralMPM to bypass expensive graph constructions ... constant runtime.") of the effectiveness on efficiency of the proposed method comes from MPM itself but not new knowledge introduced by the NeuralMPM.

**Questions:**

- For MPM simulation, the particle-to-grid(p2g) and grid-to-particle(g2p) are generally considered as expensive stages or bottleneck of the runtime performance[1]. As scatter or gather operations are more challenging for parallel computation comparing to in-place operations of grid-updates. Just wondering what is the motivation of making surrogate model for grid-update part instead of p2g or g2p when developing NeuralMPM?

- In Table 1, the proposed NeuralMPM performs much better than GNS on VariableGravity (MPM) case (14.48 vs 134 on MSE) and MultiMaterial (MPM) (9.6 vs 14.79), however worse on WaterRamps(MPM) and SandRamps(MPM). These three cases are all MPM based, what do you think the reasons why there are such as huge performance gaps between similar cases? In addition, the proposed method also show much advantages on the SPH based case (i.e., Dam Break 2D), would you mind sharing some intuition on why this is the case?

- In Inference time and memory part and Figure 6, it demonstrates that the proposed method is applied to cases with 1M (i.e., 1000k) or even 10M (i.e., 10000k) particles, while in Table 1 the model is trained on cases with less than 5k particles. Just wondering if the model are re-trained on the 1M or 10M cases or they can be trained on 5k and transfer to a 200 or even 2000 times larger case?

- For the generalization, the authors mention that the model trained on 64x64 grid size can generalize to 64 x128 without re-training. Just wondering would you mind providing a larger case until the generalization fails? It would be helpful to understand what is the limit of the generalization ability.

[1] Wang et al, A Massively Parallel and Scalable Multi-GPU Material Point Method, ACM Trans. Graph (2020)

---

> ### Author Response · Authors · 2024-11-15
>
> Thank you for your review. We answer your questions and issues to the best of our ability below. We appreciate the feedback and value constructive critique, though we find it challenging to identify actionable steps based on these arguments. Could you clarify the arguments leading to a rejection, and whether your perspective has changed given our explanations below?
>
> -----
>
> Now, to answer the concerns you raise in weakness section:
>
> 1) Our method aims to emulate and accelerate existing simulation approaches by learning directly from data, rather than achieving the most accurate fluid simulation. While NeuralMPM is inspired by and incorporates elements of the MPM framework, it is data-driven, allowing it to effectively adapt beyond the usual MPM domains. This includes emulating simulations for incompressible fluids and SPH data (as shown in Table 1). That is, with adequate training data, the inductive bias introduced by MPM can be "overwritten".  Additionally, we demonstrate that NeuralMPM outperforms fully Lagrangian data-driven methods like GNS in accuracy and stability.
>
> 2) We acknowledge throughout the paper that our method is based on the classic MPM and therefore benefits from its advantages (L52-L64). In L519-521 we do not claim that those advantages are unique to NeuralMPM or introduced by us, rather, we are claiming that NeuralMPM has those advantages over the baselines we compared against. We consider inheriting the advantages of MPM a strength, compared to other data-driven methods.
>
> ---------
> Regarding your questions:
>
> 1) The motivation stems from the computational limitations of surrogating gather operations. Any ML surrogate would have at least O(N) complexity, the same as scatter and gather operations. A tree based GNN surrogate, for instance, would require O(N log N) computations to build the graph, without taking into account additional operations, the costly training procedure, or the extra error introduced by the surrogate. [3-4] are examples of this.
>
> 2) On WaterRamps and SandRamps, there is a performance gap between NeuralMPM and GNS in terms of the MSE, but NeuralMPM has better EMD. We believe that the MSE gap is due to the GNS hyperparameter tuning, and that it can be reduced with a more extensive hyperparameter search for NeuralMPM. In VariableGravity, we used both the original GNS hyperparameters and ran an extensive hyperparameter search (60 GPU days, L255) to optimise GNS performance and ensure fair comparisons, while for Dam Break 2D we used the model provided by LagrangeBench [2], but retrained it for 20 million steps (like GNS) instead of the provided checkpoint, trained for 500,000 steps. For the MultiMaterial case, the sharp boundaries between materials are challenging to model in free space but are more naturally represented on a grid. Our intuition for the better convergence of NeuralMPM  is that the combination of the voxelization and an appropriate neural architecture (UNet, FNO...) allows the model to easily capture global structures and interactions, instead of having to rely on numerous message passing steps (L107-L110). It also frees model capacity as the spatial structure is presented in a more compact way and the model does not have to infer it from the point cloud directly (L57-L62).
>
> 3) NeuralMPM can simulate a larger number of particles without retraining. We present such results in Section 4.3 (L470 - L478). In particular, we evaluate a model trained on WaterRamps (2.3k particles approximately and 600 timesteps) on WaterDrop-XL (7.1k particles approximately and 1000 timesteps). Some of the WaterDrop-XL simulations have up to 10k particles, an almost 5x increase in the number of particles. In Fig. 6, we limit-test the speed and memory of both NeuralMPM and the baselines, without retraining.
>
> We will add a section in the appendix with generalization to progressively larger grids, until it fails.
>
> -----
> We thank you again for your review. We propose to improve our generalization experiments with the larger grid until it fails. With the perspective provided by our answers, could you provide additional actionable input, or elaborate on how you have arrived at the decision to reject?
>
> ----
>
>
> References
>
> [1] Wang et al, A Massively Parallel and Scalable Multi-GPU Material Point Method, ACM Trans. Graph (2020)
>
> [2] Toshev et al. (2024). Lagrangebench: A lagrangian fluid mechanics benchmarking suite. Advances in Neural Information Processing Systems, 36. https://github.com/tumaer/lagrangebench
>
> [3] Hao, Z., Ying, C., Wang, Z., Su, H., Dong, Y., Liu, S., Cheng, Z., Zhu, J., & Song, J. (2023). GNOT: A General Neural Operator Transformer for Operator Learning. ArXiv, abs/2302.14376.
>
> [4] Li, Z., Kovachki, N.B., Choy, C., Li, B., Kossaifi, J., Otta, S.P., Nabian, M.A., Stadler, M., Hundt, C., Azizzadenesheli, K., & Anandkumar, A. (2023). Geometry-Informed Neural Operator for Large-Scale 3D PDEs. ArXiv, abs/2309.00583.

---

> > ### Comment · Reviewer_sSsx · 2024-11-20
> > **Response**
> >
> > Thanks for the reply.
> >
> > I am not convinced by the authors' clarification regarding to the weaknesses, as the claim is somehow contradictory. The author mentions that "Our method aims to emulate and **accelerate** existing simulation... **rather than achieving the most accurate fluid simulation**". However, the following statements argue that "with adequate training data, the **inductive bias introduced by MPM can be 'overwritten'**", which indicates the proposed method can actually outperform MPM with regard to accuracy given good SPH data. Just wondering is the aim of the proposed method to accelerate or enhance the existing simulation method to overcome their inherent limitations? Or the proposed the method can do both?
> >
> >
> > If it is the previous one (i.e., accelerate), then the limitation of the MPM is inherited as I mentioned in the weakness section. If it is the latter one (i.e., enhance), then there should be more experiments to support this claim. For example, given a good plasma dataset, can the proposed method use MPM to simulate the behavior of plasma? Obviously, there is no sufficient experiments for the latter point (i.e., enhance). As a result, I don' think the authors can escape the scope of MPM if they solely would like to accelerate the simulation. Thus, the technical contribution of this paper is limited under this context. That makes me arrive at the decision to reject.
> >
> >
> > For more actionable inputs:
> > - One key claim of the proposed method is that it strongly outperforms all three methods (Line 441 to 442) regarding to the inference time and memory. In Figure 6, the test case has 30 million particles as described in Line 430, which is far more large than the training and other test cases. Is it a 3D case or 2D case？Just wondering would you minding showing some qualitative or visual results of the case in Figure 6 ? The reason is that the visual results can help check whether the simulations are still converged. There will be a substantial performance gap between converged and diverged simulation. Also, would you mind providing metrics regarding to accuracy i.e., MSE and EMD for this case?
> >
> > - It seems that the author still didn't introduce which trained model is used for the inference and memory test as I list in the question. The reason for this question is that the training data mentioned is less than 5K however the test cases can be 30 million. If the model can be directly applied to the 30 million without re-training, it indicates the model has a very strong generalization ability. Would you help clarify this point please?
> >
> > - For the generalization experiment (Figure 7), the author mentions that "No ground truth is displayed as Sanchez-Gonzalez et al. (2020) provide no information about the data generation."However, as the authors can run Taichi-MPM, is there any reason not using this simulator to generate data with ground truth for testing? It will be more convincing if test cases with ground truth can be provided.

---

> ### Author Response · Authors · 2024-11-20
>
> Thank you for your response, and for clarifying the reasons behind your decision. We appreciate the concrete feedback you provided and we will clarify some points and address first the weaknesses, then the rest.
>
> -----
>
> We are sorry for the confusion. In the statement "with adequate training data, the inductive bias introduced by MPM can be 'overwritten'", we wanted to stress that our method is not limited to the domains of traditional MPM, but can be applied to other domains as well, although we expect it to be especially well suited to traditional MPM domains. Regarding "which indicates the proposed method can actually outperform MPM with regard to accuracy given good SPH data," we clarify that MPM and SPH are fundamentally different dynamical models, each with unique strengths and weaknesses. Our point was that, while MPM cannot replicate SPH-like simulations due to its construction, our method can, when trained on SPH data.
>
> In Fig. 6, we show that our method is faster than the MPM ground truth (acceleration). With regards to the enhancement, we do not claim to be superior to traditional methods, we claim to be faster, and to be able to solve inverse problems (Section 4.4) which we consider an enhancement. Importantly, we claim and show to be faster, more accurate, and have better memory scaling than other data-driven methods.
>
> We will reformulate our goal explicitly: "Our method aims to emulate and accelerate existing simulation approaches, while being superior in terms of speed, memory, and accuracy to existing data-driven methods, and being able to solve inverse problems.". That is, our main contribution is the improvement upon existing data-driven baselines.
>
> ----
>
> (Points 1 and 2) We support our claims for generalization from 2.3k particles to ~8k in Section 4.3 and Fig. 7. We do not claim that our method generalized from 5k to 30M particles, nor do any of the baselines. In Fig. 6 we tested the throughput and memory of the untrained models (baselines and NeuralMPM). We did not see any difference between the converged models and the unconverged ones, for neither the baselines nor NeuralMPM, as the particles are always kept within the domain by construction (no large numbers or exploding quantities). We will make this explicit in the revised version.
>
> (Point 3) We did not run the same simulator as Sanchez-Gonzalez et al. (2020), as it has been unmaintained for years and we were not able to compile it (<https://github.com/yuanming-hu/taichi_mpm>). We did have access to a subsequent version, but we were not able to replicate the dataset. We note that there is a video in the supplementary material showing various realistic simulations. However, as further evidence and as told by the other reviewers, we will create ground-truth data and show it explicitly using an SPH simulator.
>
> ----
> We hope to have addressed your concerns, and will add ground truth for generalization within the rebuttal period. We kindly ask you to reconsider your decision in light of our responses, especially the improvements over data-driven methods.

---

> ### Comment · Reviewer_sSsx · 2024-11-25
> **Further response**
>
> Thanks for the reply. I appreciate the authors clarify the motivation and claims.  However, some points for supporting the claims are not solid enough: **1) accelerate existing simulation approaches**, in Figure 6, the authors compare their untrained model to the traditional MPM, it is not convincing since it is unclear whether the trained model would diverge the simulation or not.  Diverged/undiverged simulations have very different particles distributions, which affects the performance dramatically. Even the simulation is not diverged, the performance between a highly dynamic and quasi-stationary simulation is also very different. The authors would better compare their trained and well-behaved (i.e., can output reasonable simulation results) model to traditional simulators. Otherwise it is not a fair comparison as it is actually just running the low level CUDA operators but not the proposed method in a simulation context.  **2) being able to solve inverse problems**The ability to solve the inverse problem is not an enhancement as there are already efficient differentiable MPM solvers several years ago (difftaichi [1]) there are also bunch of downstream applications using the diff mpm such as plasticinlab[2], PAC-nerf[3], PhysGaussian[4] and many more. The cases shown in these previous work are much complex than the one shown in the paper. Therefore it is not convincing to state this point as a claim without any comparisons to existing differentiable MPM solvers. I suggest that the authors revise the current manuscript and submit to another venue.
>
>
> [1] Hu et al, Differentiable Programming for Physical Simulation (ICLR 2021)
>
> [2] Huang et al, Plastcinelab: A soft-body Manipulation Benchmark with Differentiable Physics (ICLR 2022)
>
> [3] Li et al, PAC-NeRF: Physics Augmented Continuum Neural Radiance Fields for Geometry-Agnostic System Identification (ICLR 2023)
>
> [4] Xie et al, Physics-Integrated 3D Gaussians for Generative Dynamics (CVPR 2023)

---

> > ### Author Response · Authors · 2024-11-27
> >
> > Thank you for engaging with us in the discussion!
> >
> > The motivation behind the neural emulation field is the acceleration of existing simulations, the ability to solve inverse problems, and to account for unknown dynamics. This is a goal our method shares with both baselines, as well as many other papers.
> >
> > We once again highlight that the main goal of our work is to compare against other neural emulation frameworks and that our main claim is that we achieve similar or superior accuracy in 6% of the training time (15h compared to 10 days), have a 5x-10x faster inference time, and use 10x-100x less memory. This is our main contribution. To avoid any possible confusion, we will contact the AC and SAC to change the paper name to "A Neural Material Point Method for Particle-based *Emulation*".
> >
> > Further, NeuralMPM is faster than the MPM baseline also in the small simulation regime, where we used trained models whose rollouts and ground truth are given in the paper (MSE and EMD in Table 1, rollouts in supplementary videos and Fig. 12). In that regime the model is converged and all the steps are meaningful.
> > Moreover, in Fig. 6 we have measured the FPS, and not the time it takes to simulate a given time interval. In reality, the neural emulators (including NeuralMPM) use a much larger step size ($2.5$ms) than the ground truth simulator ($0.2$ms), which diverges if a large step size is used.
> > This means the real speedup of neural emulators is much larger than shown in Fig. 6, with both baselines likely exceeding the MPM simulator. NeuralMPM is even faster, as it generates frames faster and has a larger step size.
> >
> > While differentiable simulators exist they are not the norm and require careful construction. Neural emulators in general and our method in particular are differentiable out of the box. The works you point out have impressive results, but we once again stress that our main goal is to compare against other neural emulators and that our main contribution is the speed, memory, and accuracy improvements over these. We do not claim that there are no differentiable simulators, nor that we are better than them, but a lot of simulators are not differentiable, and our method is.

---

### Author Response · Authors · 2024-11-15
**General comment**

We thank all the reviewers for their reviews and feedback. We have addressed their questions and concerns, and will provide a revision with the proposed changes within the rebuttal period. Reviewers **sSsx,** **w7nQ,** and **DPie** found the paper clear, well written, and informative. **w7nQ** highlights that the method is demonstrated to be more efficient than traditional MPM and the other data-driven baselines in terms of memory, runtime, training time, and accuracy. They also remark that it allows for the use of very uncomplicated processors, and that it can solve inverse problems. **DPie** highlights the strengths of leveraging the MPM framework in a data-driven setting, the inductive biases it provides, and data efficiency of NeuralMPM.

**DPie** asked for 1) plots and visual comparisons with the baselines, 2) that we elaborate on the performance of GNS, 3) a clarification and more proof of the claim of invariance with respect to the number of particles, and 4) reformulation of the abstract and conclusions. **w7nQ** raised concerns about the long term stability of the rollouts, the generalization to different grid resolutions, and the scaling to 3D. **sSsx** elaborates on a number of perceived weaknesses, but does not provide actionable steps. We have attempted to address those weaknesses with proposed changes, and asked for further input. Similarly, **GJwq** lists perceived weaknesses without clear actionable steps. We have answered their questions, proposed changes to address the perceived weaknesses, and asked for further input.

Importantly, we want to clarify that our work is not an extension of the MPM simulation framework, nor is it a direct competitor. NeuralMPM is a fully data-driven neural emulator that uses some of the ideas of MPM to solve problems that appear within the neural emulator literature. That is, our primary goal is not physical realism (as you would expect in physics-driven or hybrid models) but the emulation of existing simulators. As such, we benchmark NeuralMPM against other neural emulators only, and showcase its advantages across a wide range of experiments (Table 1, Fig. 4). Neural emulators complement classic simulators in that they can be learned from data, avoiding the very complex process of crafting a simulator, are differentiable out of the box, which is extremely useful for inverse problems, and can have better memory and speed performance. We demonstrate that last point with a comparison in terms of memory consumption and simulation speed to Taichi-MPM, the ground truth simulator that was used to generate the data (Fig. 6).

We ask the reviewers, in light of this clarified perspective, the answers provided, and suggested changes (which will be included in a revision within the rebuttal period), to reevaluate their scores.


**A note on the formatting of our answers: They should be read as a Q/A, where each paragraph is addressing a question or concern from the relevant section in the reviewer's response.**

---

> ### Author Response · Authors · 2024-11-27
>
> Dear reviewers and area chair,
>
> We thank all the reviewers for the discussion so far. We have uploaded the revised version of the paper, with changes highlighted in red. In the (updated) supplementary materials, you will find new rollout videos showing the baselines, NeuralMPM, and the ground truth for all experiments, as well as the generalization rollouts for NeuralMPM. Due to size constraints, some are in the supplementary materials and the rest are in this anonymous drive <https://drive.google.com/drive/folders/1lj8a-iHoAXOyT3HFl5mXa0Bc5_nHbdZb>. This will help the comparison and support the accuracy claims of NeuralMPM in Table 1. A non-exhaustive summary of the changes we made in the revised version is as follows. Further details are given in the response to the reviews below.
>
>  1. We propose changing the name of the paper to "A Neural Material Point Method for Particle-based *Emulation*" to better reflect our contribution.
>  2. Reworded the abstract and conclusion to be more quantitative.
>  3. Updated Fig. 4 and added similar figures for the other datasets in the appendix.
>  4. Updated Fig. 6 caption.
>  5. Added Fig. 9 and and Fig. 10 in the appendix.
>  6. Added videos of all rollouts in the supplementary materials.
>  7. Reworded the invariance claim to be about the processor, and added the linear scaling of p2g and g2p.
>  8. Added an extra comparison in the appendix against a different implementation of GNS and SEGNN (Brandstetter et al. 2021), as provided by LagrangeBench (Toshev et al. 2024). Our conclusions remain the same.
>  9. Added Figure 11 in the appendix to show that NeuralMPM performs better than GNS at the WaterDrop-XL generalization task, despite using a model with a slightly lower MSE.
>
> In addition, given how the discussion phase is unfolding, we feel compelled to address, once again, a fundamental misalignment in how our work is being evaluated. NeuralMPM should be assessed as a data-driven emulator that learns from simulation data, not as a physics engine competitor. This distinction is crucial. We strongly emphasize that our main contribution and claim is not superiority to traditional numerical methods, nor an improved version of MPM. We do not claim that in the paper. Rather, we claim and show to strongly outperform the neural emulator baselines. Previous works in this space (GNS, DMCF) were evaluated based on their ability to learn from and reproduce simulation data, without requirements to validate against multiple simulators or justify every observed physical pattern. They showed results on similar datasets with similar evaluation protocols. Our work substantially improves upon these methods in terms of:
>
> - Training speed: 15h compared to 10 days (i.e. 6% of the training time)
> - Inference speed: 5x-10x faster
> - Memory usage: 10x-100x less
> - Accuracy: superior EMD for all datasets, superior MSE for all datasets except 2, where it is very close.
> - Multi-material interaction (MultiMaterial dataset, see videos in the supplementary material)
> - Inverse problems (while both GNS and DMCF are in principle differentiable, they do not show results on inverse problems)
>
> The additional requirements being suggested - such as validating against multiple simulator types, going 3D, justifying every observed flow pattern, asking for physical realism beyond the capabilities of the underlying simulator (look at the ground truth of some of the videos in the supplementary material where particles stick to the walls, which was simulated using the well established Taichi-MPM) or demanding reimplementations of previous data-driven emulators over official codebases - go well beyond what was expected of previous work in this space and create unreasonable barriers for research progress (see e.g <https://openreview.net/forum?id=6niwHlzh10U> for the reviews of DMCF). Evaluating NeuralMPM as a physics engine replacement is not only unfair but also misses the key contributions of our work.
>
> As discussed all along, we are happy to improve clarity around our claims and positioning further -- and we thank the reviewers for their feedback in this regard -- but ask that the work be evaluated within its intended context as a machine learning contribution advancing the state-of-the-art in neural simulation emulation.
>
> The authors.

---

> > ### Comment · Reviewer_DPie · 2024-11-28
> >
> > You state that your primary goal is not physical realism (as you would expect in physics-driven or hybrid models) but the emulation of existing simulators. Isn't a simulator, which is meant to simulate the physical behaviour of particles, supposed to be physically realistic? Wouldn't you call an unrealistic simulator a bad simulator?

---

> ### Author Response · Authors · 2024-11-29
>
> The drive folder now includes updated videos of the WaterDrop-XL rollouts, showing the ground truth, NeuralMPM, and GNS for comparison. The GNS rollouts are worse, supporting the results from Fig. 11. Note that both modles, GNS and NeuralMPM, are the  ones reported in Table 1 on WaterRamps. GNS has slightly better MSE than NeuralMPM on that dataset. Despite that, NeuralMPM generalized better.

---

### Note · Authors · 2024-12-02

**Comment:**

We appreciate the time the reviewers have dedicated to their reviews and our answers. However, due to the likely negative outcome we have decided to withdraw our paper.

We find that the reviews fundamentally misalign with the intended scope and contributions of our paper. Our work focuses on advancing neural emulation frameworks, specifically addressing challenges of training efficiency, inference time, and memory scaling in comparison to existing neural emulators. We have shown strong improvements in all those metrics against GNS and DMCF, using the implementations of the original authors in both cases.

Throughout the review process, we clarified this perspective and provided substantial updates and evidence, including extended results, new visualizations, complete comparisons, and clarified claims. Despite our efforts, the evaluation criteria and expectations of the reviewers appear inconsistent with those applied to similar works in the field, and their concerns extend beyond the paper's scope and stated contributions.

As a result, we believe continuing the review process under these circumstances is not constructive. We intend to submit a revised version of our work to a venue more aligned with its contributions and objectives.

Thank you for your participation.

**Withdrawal Confirmation:**

I have read and agree with the venue's withdrawal policy on behalf of myself and my co-authors.